# Mechanism and function of DNA replication-independent DNA-protein crosslink repair via the SUMO-RNF4 pathway

Julio C Y Liu[1,†] [ID], Ulrike Kühbacher[1,†] [ID], Nicolai B Larsen[1], Nikoline Borgermann[1], Dimitriya H Garvanska[1], Ivo A Hendriks[1] [ID], Leena Ackermann[1], Peter Haahr[2] [ID], Irene Gallina[1], Claire Guérillon[1] [ID], Emma Branigan[3], Ronald T Hay[3] [ID], Yoshiaki Azuma[4] [ID], Michael Lund Nielsen[1], Julien P Duxin[1,*] [ID] & Niels Mailand[1,5,**] [ID]

## Abstract

DNA-protein crosslinks (DPCs) obstruct essential DNA transactions, posing a serious threat to genome stability and functionality. DPCs are proteolytically processed in a ubiquitin- and DNA replication-dependent manner by SPRTN and the proteasome but can also be resolved via targeted SUMOylation. However, the mechanistic basis of SUMO-mediated DPC resolution and its interplay with replication-coupled DPC repair remain unclear. Here, we show that the SUMO-targeted ubiquitin ligase RNF4 defines a major pathway for ubiquitylation and proteasomal clearance of SUMOylated DPCs in the absence of DNA replication. Importantly, SUMO modifications of DPCs neither stimulate nor inhibit their rapid DNA replication-coupled proteolysis. Instead, DPC SUMOylation provides a critical salvage mechanism to remove DPCs formed after DNA replication, as DPCs on duplex DNA do not activate interphase DNA damage checkpoints. Consequently, in the absence of the SUMO-RNF4 pathway cells are able to enter mitosis with a high load of unresolved DPCs, leading to defective chromosome segregation and cell death. Collectively, these findings provide mechanistic insights into SUMO-driven pathways underlying replication-independent DPC resolution and highlight their critical importance in maintaining chromosome stability and cellular fitness.

**Keywords** DNA repair; DNA-protein crosslinks; genome stability; SUMO; ubiquitin

**Subject Categories** DNA Replication, Recombination & Repair; Post-translational Modifications & Proteolysis

**The EMBO Journal (2021) 40: e107413**

## Introduction

Proteins acting on or residing in proximity to DNA can become covalently trapped, giving rise to DNA-protein crosslinks (DPCs). These bulky lesions can undermine the integrity and functionality of the genome, as DPCs interfere with essential DNA-associated transactions including DNA replication and transcription. In addition to the accidental trapping of enzymes forming covalent intermediates with DNA during catalysis, DPCs can be generated with varying efficacy by a wide range of both endogenous and exogenous agents including reactive aldehydes and commonly used chemotherapeutic drugs such as topoisomerase inhibitors and cisplatin (Ide *et al*, 2011; Tretyakova *et al*, 2015; Kühbacher & Duxin, 2020). The action of DPC-inducing genotoxins and the stochastic trapping of proteins on DNA together underlie an enormous diversity of DPCs, which differ with respect to at least three basic parameters: the identity of the crosslinked protein, the chemical nature of the covalent linkage formed between DNA and the protein adduct, and the structure of the DNA flanking the DPC. Faced with the formidable challenge of recognizing and repairing a highly diverse range of DPCs, eukaryotic cells possess a versatile toolbox of DPC resolution factors and mechanisms, whose complexity has become apparent through extensive research efforts over the past few years. A central strategy for resolving DPCs conserved from yeast to humans involves DNA replication-coupled mechanisms targeting the protein component of the DPC. In particular, the SprT-type metalloproteases Wss1 (yeast) and SPRTN (higher eukaryotes) have firmly established key roles as replication-coupled DPC repair factors that proteolytically process covalent protein adducts to facilitate replication past the lesions (Duxin *et al*, 2014; Stingele *et al*, 2014; Stingele *et al*, 2016; Vaz *et al*, 2016; Larsen *et al*, 2019). While SPRTN can degrade a wide

1   Protein Signaling Program, Novo Nordisk Foundation Center for Protein Research, University of Copenhagen, Copenhagen, Denmark
2   Netherlands Cancer Institute, Amsterdam, The Netherlands
3   Centre for Gene Regulation and Expression, School of Life Sciences, University of Dundee, Dundee, UK
4   Department of Molecular Biosciences, University of Kansas, Lawrence, KS, USA
5   Center for Chromosome Stability, Department of Cellular and Molecular Medicine, University of Copenhagen, Copenhagen, Denmark
    *Corresponding author. Tel: +45 93565571; E-mail: julien.duxin@cpr.ku.dk
    **Corresponding author. Tel: +45 35325023; E-mail: niels.mailand@cpr.ku.dk
    †These authors contributed equally to this work

range of different protein substrates and could in principle act on replisome-associated proteins in a non-specific manner with potentially deleterious consequences, a range of elegant regulatory mechanisms ensure exquisite selectivity of SPRTN for cleaving DPCs. These include a ubiquitylation–acetylation switch that controls SPRTN chromatin association and dual DNA-binding domains that specifically target its protease activity to the junction between single- and double-stranded DNA (ssDNA and dsDNA) formed when nascent DNA strands stall at a DPC (Stingele *et al*, 2016; Larsen *et al*, 2019; Li *et al*, 2019; Huang *et al*, 2020; Reinking *et al*, 2020). In addition to SPRTN, we recently demonstrated that the proteasome is targeted to ubiquitylated DPCs and promotes an alternative SPRTN-independent protease-driven DPC-processing pathway during DNA replication (Larsen *et al*, 2019). Moreover, recent evidence suggests that additional proteases, including DDI1/2 and FAM111A, may also contribute to DNA replication-dependent DPC processing (Kojima *et al*, 2020; Serbyn *et al*, 2020). Collectively, these mechanisms ensure highly efficient clearance of DPCs encountered during DNA replication to mitigate their adverse impact on genome duplication. Reflecting the physiological importance of such mechanisms, *SPRTN* mutations in humans give rise to a rare syndrome characterized by premature ageing and increased cancer susceptibility (Lessel *et al*, 2014).

Despite the efficiency of DNA replication-coupled DPC resolution pathways, these mechanisms are unlikely to offer full protection from the collective threat posed by DPCs, which can arise during any stage of the cell cycle. For instance, DPCs formed after DNA replication may be out of reach for DNA replication-coupled repair mechanisms, and it is conceivable that an inability to resolve such lesions could endanger subsequent faithful chromosome segregation. Likewise, non-dividing cells, particularly long-lived cells such as neurons, likely require DPC recognition and resolution mechanisms to avoid progressive accumulation of these lesions, which could undermine long-term cellular fitness. Indeed, previous findings raised the possibility that cells are able to resolve DPCs in a manner that is uncoupled from the replication fork and that the small ubiquitin-like modifier SUMO may have a key role in this process. Specifically, we recently demonstrated that DNMT1 DPCs, which can be generated post-replicatively in genomic DNA by treatment with the cytosine analogue 5-aza-2′-deoxycytidine (5-azadC) and are not accompanied by DNA strand breaks, are not resolved by SPRTN but undergo robust and direct SUMOylation, a modification that is essential for the subsequent removal of these lesions (Borgermann *et al*, 2019). Similarly, both Topoisomerase 1 and 2 (TOP1 and TOP2) cleavage complexes, which are flanked by DNA breaks, are modified by SUMOylation to promote their clearance, and available evidence indicates that the resolution of this type of DPC occurs efficiently outside S phase (Mao *et al*, 2000; Mo *et al*, 2002; Heideker *et al*, 2011; Steinacher *et al*, 2013; Schellenberg *et al*, 2017; Sharma *et al*, 2017; Wei *et al*, 2017; Sun *et al*, 2020). We and others recently showed that the SprT protease ACRC (also known as GCNA) is a putative DPC protease that interacts with SUMOylated DPCs via tandem SUMO-interacting motifs (SIMs) (Borgermann *et al*, 2019; Bhargava *et al*, 2020; Dokshin *et al*, 2020). However, the expression and function of ACRC/GCNA appear to be largely confined to germline cells (Carmell *et al*, 2016). Thus, how SUMOylation of DPCs, in particular those residing at uninterrupted duplex DNA, drives lesion processing and resolution, its interrelation with DNA replication-

coupled DPC repair and relative biological importance for preventing genomic instability remain unclear.

In this study, we addressed these important questions. We demonstrate that DPCs on duplex DNA are SUMOylated in the absence of DNA replication and subsequently targeted for proteasomal degradation by the SUMO-targeted ubiquitin ligase (STUbL) RNF4. SUMOylation of DPCs neither stimulates nor inhibits their DNA replication-dependent proteolysis. Instead, the SUMO-RNF4 pathway provides an independent salvage route to remove DPCs formed or persisting after DNA replication. Notably, we show that cells harbouring DPCs on duplex DNA do not activate the interphase DNA damage checkpoint and enter mitosis even in the presence of a high load of protein adducts, leading to defective chromosome segregation and loss of viability. Together, these findings reveal the mechanistic basis of SUMO-driven, DNA replication-independent DPC resolution and underscore its critical importance in safeguarding genome integrity.

# Results

## RNF4 promotes resolution of SUMO-modified DPCs in duplex DNA

To understand the mechanistic basis of how SUMOylation promotes removal of DPCs, we first monitored the resolution kinetics for DNMT1 DPCs that are formed in the wake of the replication fork following 5-azadC incorporation into genomic DNA during replication. To this aim, cells were synchronized in early S phase, exposed to a brief (30-min) pulse of 5-azadC in order to ensure even incorporation of this nucleotide analogue into genomic DNA of all cells, and analysed at different time points (Fig 1A). Methylation of 5-azadC by DNMT-type methyltransferases, in particular DNMT1 that methylates CpG motifs in newly replicated DNA to maintain DNA methylation patterns, leads to their covalent crosslinking to DNA. Cell fractionation revealed that the SUMO-dependent modification and resolution of trapped DNMT1 molecules proceeded with rapid kinetics, as the bulk of SUMOylated DNMT1 DPCs were cleared from chromatin within 3-4 h after 5-azadC treatment (Fig 1A). This was accompanied by a complete loss of the soluble pool of DNMT1 molecules (Fig 1A), suggesting that crosslinked DNMT1 molecules are degraded following their entrapment on chromatin. Complementing this analysis, we used an automated high throughput imaging approach to quantify DNMT1 DPC resolution kinetics across large populations of individual cells, using detergent-resistant DNMT1 nuclear foci as a readout for endogenous DNMT1 molecules immobilized on chromatin upon 5-azadC treatment (Borgermann *et al*, 2019). In this assay, trapped DNMT1 molecules were largely cleared from chromatin by 3 h after 5-azadC removal, paralleling DNMT1 DPC resolution kinetics observed by chromatin fractionation (Figs 1A and B, and EV1A). Blocking DNMT1 DPC SUMOylation by means of the SUMO E1 inhibitor ML-792 (SUMOi; (He *et al*, 2017)) suppressed the removal of DNMT1 DPCs as we previously reported (Fig 1C) (Borgermann *et al*, 2019). Likewise, inhibiting the proteasome or overall ubiquitylation also impaired the removal of DNMT1 DPCs (Figs 1C and EV1B), and the formation of DNMT1 DPCs was accompanied by their modification with K48-linked ubiquitin chains (Fig 1D). These observations suggested the involvement of STUbL activity in promoting ubiquitylation of DNMT1 DPCs downstream of

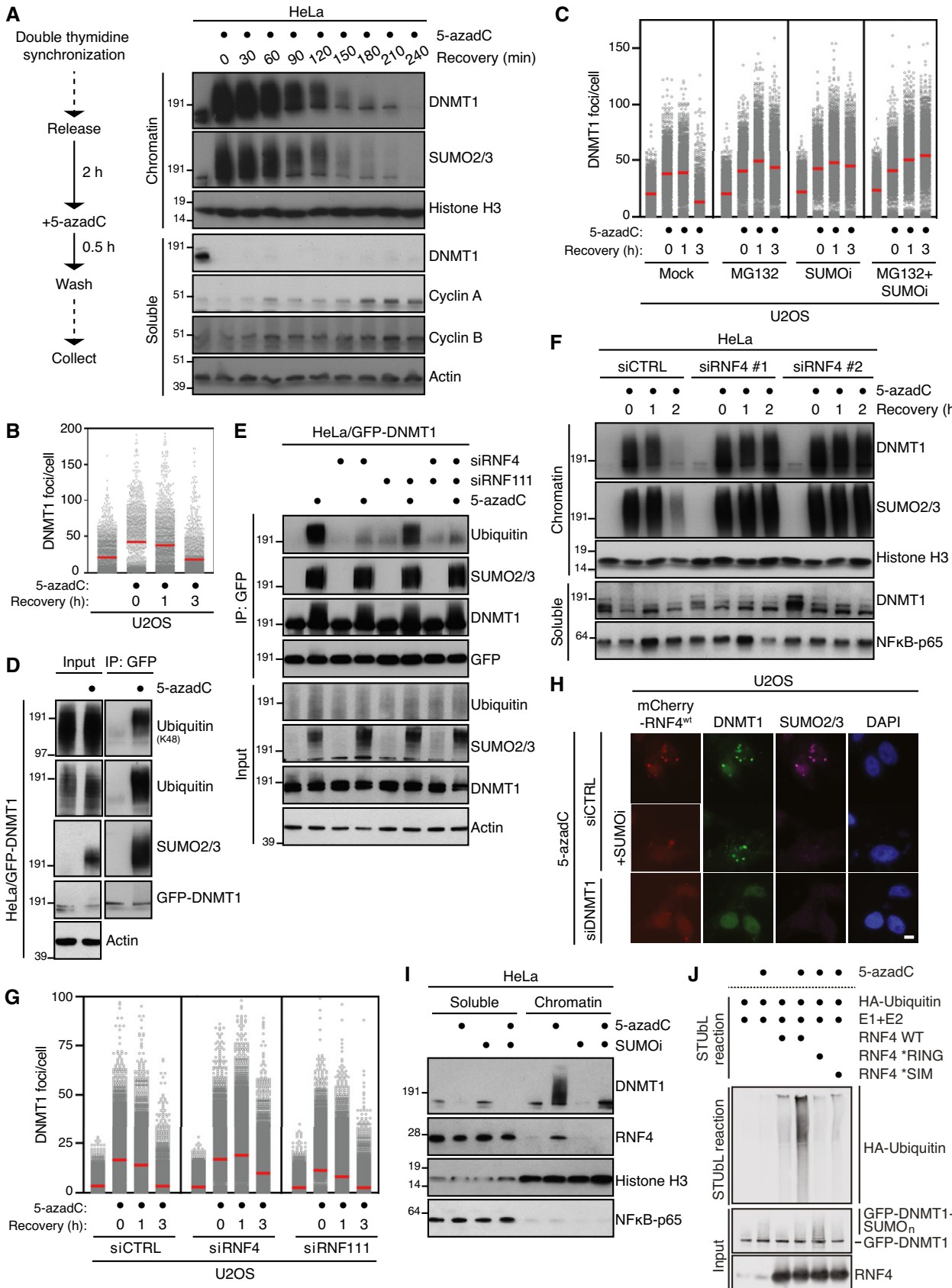

**Figure 1.**

**Figure 1. RNF4 promotes resolution of SUMO-modified DNMT1 DPCs.**

A  HeLa cells released from double thymidine synchronization in early S phase were treated with 5-azadC for 30 min, washed and collected at the indicated times. Soluble and chromatin-enriched fractions were immunoblotted with indicated antibodies.

B  U2OS cells treated as in (A) were pre-extracted and immunostained with DNMT1 antibody. DNMT1 foci formation was analysed by quantitative image-based cytometry (QIBC) (red bars, mean; > 1,500 cells analysed per condition). Data are representative of three independent experiments. See also Fig EV1A.

C  As in (B), except that cells were pre-treated with the proteasome inhibitor MG132 and/or SUMOi for 30 and 15 min, respectively, before exposure to 5-azadC (red bars, mean; > 1,900 cells analysed per condition). Data are representative of three independent experiments.

D  HeLa cells stably expressing GFP-DNMT1 were treated or not with 5-azadC for 30 min, collected and subjected to GFP immunoprecipitation under denaturing conditions, and immunoblotted with indicated antibodies.

E  HeLa/GFP-DNMT1 cells transfected with indicated siRNAs were processed as in (D).

F  Immunoblot analysis of soluble and chromatin-enriched fractions of HeLa cells transfected with indicated siRNAs, left untreated or exposed to 5-azadC for 30 min and collected at the indicated times.

G  U2OS cells transfected with indicated siRNAs were treated with 5-azadC for 30 min and processed for QIBC analysis of DNMT1 foci counts as in (B) (red bars, mean; > 5,600 cells analysed per condition). Data are representative of three independent experiments.

H  Representative images of U2OS cells transfected with indicated siRNAs followed by mCherry-RNF4 expression plasmid. Cells were treated with 5-azadC in the presence or absence of SUMOi, fixed 2 h later, pre-extracted and co-immunostained with DNMT1 and SUMO2/3 antibodies. Scale bar, 10 μm.

I  Immunoblot analysis of soluble and chromatin-enriched fractions of HeLa cells treated with 5-azadC and/or SUMOi as indicated.

J  GFP-tagged DNMT1 from extracts of HeLa/GFP-DNMT1 cells treated or not with 5-azadC was immobilized on GFP-Trap agarose, subjected to stringent washing to remove proteins non-covalently bound to GFP-DNMT1 and incubated with recombinant HA-ubiquitin, E1 and E2 (UbcH5a) enzymes and RNF4 proteins (STUbL reaction) at 37°C for 1 h. Samples were then subjected to immunoblotting to assay for RNF4-dependent STUbL activity towards GFP-DNMT1.

their SUMOylation. Knockdown of the known human STUbLs, RNF4 and RNF111, revealed that RNF4 but not RNF111 was required for 5-azadC-induced ubiquitylation of DNMT1, whereas it was dispensable for DNMT1 SUMOylation (Figs 1E and EV1C and D). Consistently, depletion of RNF4 but not RNF111 impaired the timely clearance of 5-azadC-induced DNMT1 DPCs (Figs 1F and G, and EV1E–H). In agreement with a role of RNF4 in promoting resolution of SUMOylated DNMT1 DPCs, RNF4 was recruited to DNMT1 DPC sites and accumulated on chromatin upon 5-azadC-induced DNMT1 DPC formation in a SUMO-dependent manner (Fig 1H and I). Moreover, wild-type (WT) RNF4 but not catalytically inactive or SIM-deficient mutants promoted ubiquitylation of SUMOylated DNMT1 isolated from 5-azadC- but not mock-treated cells *in vitro*, providing direct evidence that SUMOylation of DNMT1 renders it susceptible to modification by RNF4 STUbL activity (Fig 1J). We previously showed that while DNMT1 is the main cellular target of 5-azadC-induced SUMOylation, a small range of additional proteins including known DNMT1-binding factors also display increased SUMOylation upon DNMT1 DPC formation (Borgermann *et al*, 2019). Because loss of RNF4 led to a marked delay in reversing overall 5-azadC-induced chromatin SUMOylation (Figs 1F and EV1E and G), it is likely that RNF4 STUbL activity is not exclusively targeted to DPCs but also impacts other SUMOylated proteins at DPC sites to facilitate lesion removal and re-establishment of a normal chromatin state. Supporting a general role of RNF4 in resolving DPCs in uninterrupted duplex DNA, we found that it also underwent strong enrichment on chromatin upon treatment with the non-specific DPC inducer formaldehyde (Fig EV1I), which we previously showed triggers a robust SUMOylation response (Borgermann *et al*, 2019). Moreover, RNF4 was required for counteracting chromatin-associated SUMO2/3 foci resulting from formaldehyde-induced DPC formation (Fig EV1J and K). We conclude that RNF4-mediated ubiquitylation and subsequent proteasomal degradation defines an important pathway for SUMO-dependent resolution of DPCs. We note, however, that unlike complete inhibition of DNMT1 DPC degradation by SUMOi, loss of RNF4 only partially suppressed the removal of SUMO-modified DNMT1 DPCs (Figs 1G and EV1E and F), suggesting the existence of additional, SUMO-driven but RNF4-independent mechanisms for post-replicative DNMT1 DPC resolution.

## PIAS4 promotes DPC SUMOylation in the absence of DNA replication in *Xenopus* egg extracts

To further understand the mechanistic basis of DPC SUMOylation and its interplay with replication-coupled DPC repair, we turned to the cell-free *Xenopus* egg extract system, taking advantage of our previous observation that defined DPCs involving the HpaII methyltransferase (M.HpaII) linked to duplex plasmid DNA undergo SUMOylation when incubated in nucleoplasmic egg extracts (NPE) in the absence of DNA replication (Larsen *et al*, 2019) (Fig 2A). Under these conditions, M.HpaII DPC SUMOylation occurred progressively over a 3-h time course and was enhanced when a SUMO-fused M.HpaII protein (SUMOΔGG-M.HpaII) was crosslinked to DNA (p4xDPC$^{SUMO}$) (Fig 2B; compare lanes 1–5 to 6–10). These findings reinforce the notion that SUMOylation is a principal modification impacting DPCs recognized outside the context of DNA replication. However, for reasons addressed below, unlike DNMT1 DPCs in human cells, SUMOylated M.HpaII DPCs residing on duplex DNA did not undergo further processing by ubiquitylation in NPE, as evidenced by the ability of the SUMO protease Ulp1 to quantitatively remove all modified forms of M.HpaII (Fig 2C). To address whether the SUMO response was specific to DPCs or could also occur on proteins that are non-covalently trapped on chromatin, we monitored PARP1 modifications on sperm chromatin following DNA damage by UV treatment in the presence or absence of PARP inhibitors, which stabilize PARP1 association with DNA (Pommier *et al*, 2016; Zandarashvili *et al*, 2020). Previous studies in human cells and *Xenopus* egg extracts indicated that PARP1 is SUMOylated in a chromatin-dependent manner (Martin *et al*, 2009; Messner *et al*, 2009; Ryu *et al*, 2010; Zilio *et al*, 2013). Consistently, while a fraction of PARP1 appeared mono-SUMOylated on chromatin irrespectively of DNA damage or PARP inhibition, combining these treatments specifically induced PARP1 poly-SUMOylation (Figs 2D and EV2A and B), indicating that proteins that are non-covalently immobilized on DNA can also be targeted for poly-SUMOylation. CHROMASS analysis (Räschle *et al*, 2015) identified PIAS4 (also known as PIASγ) as a SUMO E3 ligase that was enriched on sperm chromatin upon PARP1 inhibition in the presence of DNA damage in a SUMO-dependent manner (Figs 2E and EV2B; Dataset EV1). In

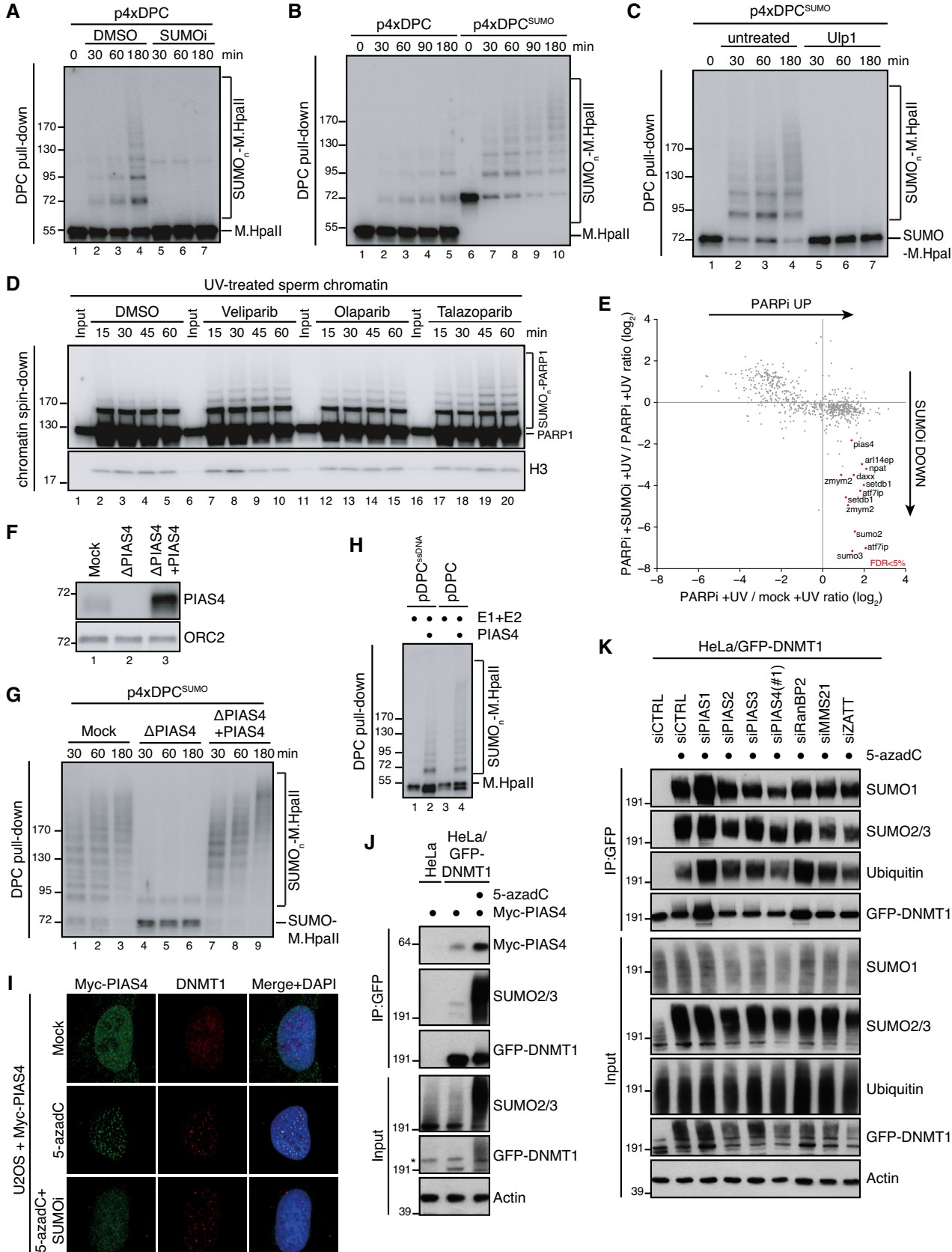

**Figure 2.**

**Figure 2.   DPC SUMOylation occurs in *Xenopus* egg extracts and is mediated by PIAS4.**

A   M.HpaII was crosslinked into a plasmid to generate p4xDPC and then incubated in nucleoplasmic egg extracts (NPE) in the presence or absence of 50 µM of SUMOi. DPC pull-down under stringent conditions was performed at the indicated time points, and the recovered samples were immunoblotted for crosslinked M.HpaII (Larsen *et al*, 2019). p4xDPC, which contains four M.HpaII DPCs, was used to increase sensitivity towards modified M.HpaII species. Identical results were obtained with plasmids containing one or four DPCs.

B   p4xDPC or p4xDPC$^{SUMO}$ (generated by crosslinking SUMOΔGG-M.HpaII) were incubated in NPE and DPC pull-down performed as in (A). Note that priming M.HpaII with SUMO stimulates rapid poly-SUMOylation of the DPC in NPE. Thus, this substrate was used in many subsequent experiments to stimulate DPC poly-SUMOylation.

C   p4xDPC$^{SUMO}$ was incubated in NPE and plasmids analysed as in (A). Following DPC pull-down, the samples were split and either left untreated or treated with the SUMO protease Ulp1.

D   Sperm chromatin treated with 2,000 J/m$^2$ of UV-C was incubated in non-replicating egg extracts in the presence or absence of the indicated PARP inhibitors (PARPi; Veliparib (50 µM), Olaparib (50 µM), Talazoparib (10 µM)). At the indicated time points, chromatin was recovered via chromatin spin-down and samples were immunoblotted with indicated antibodies.

E   Plot illustrating protein recruitment to UV-treated sperm chromatin in the presence or absence of Talazoparib and SUMOi, as determined by CHROMASS analysis (Dataset EV1). Red dots indicate the proteins that are significantly enriched on sperm chromatin in the presence of PARPi in a SUMO-dependent manner (*n* = 4 biochemical replicates; FDR < 5% corresponds to a permutation-based FDR-adjusted *q*-value of < 0.05). Note that different isoforms of the same protein (e.g. ATF7IP) can sometimes be detected.

F   NPE was either mock- or PIAS4-depleted, and recombinant xPIAS4 was added where indicated to a final concentration of 10 ng/µl. Protein samples were immunoblotted with the indicated antibodies.

G   Samples from (F) were added to p4xDPC$^{SUMO}$ for the indicated times and recovered via DPC pull-down as in (A).

H   pDPC or a plasmid containing M.HpaII crosslinked to ssDNA (pDPC$^{ssDNA}$) (Larsen *et al*, 2019) was incubated with SUMO E1 and E2 enzymes and SUMO, in the presence or absence of recombinant xPIAS4. Samples were recovered by DPC pull-down and blotted against M.HpaII as in (A).

I   Representative images of U2OS cells transfected with Myc-PIAS4 expression plasmid that were left untreated or exposed to 5-azadC in the presence or absence of SUMOi, fixed 2 h later, pre-extracted and co-immunostained with DNMT1 and Myc antibodies. Scale bar, 5 µm.

J   HeLa or HeLa/GFP-DNMT1 cells left untreated or exposed to 5-azadC for 30 min were lysed and subjected to GFP IP under stringent conditions. After extensive washing, individual IPs were incubated with an equal amount (800 µg) of whole cell lysate of HeLa cells transfected with Myc-PIAS4 expression construct (Fig EV2J), washed and immunoblotted with antibodies to Myc, SUMO2/3 and GFP. *, cross-reactive band.

K   HeLa/GFP-DNMT1 cells transfected with previously validated siRNAs targeting established SUMO E3 ligases (Fig EV2L and Methods section) were treated with 5-azadC for 30 min, collected and subjected to GFP immunoprecipitation under denaturing conditions, and immunoblotted with antibodies to SUMO1, SUMO2/3, ubiquitin and GFP.

line with this, immunodepletion of PIAS4 from NPE completely abrogated SUMOylation of M.HpaII DPCs (Figs 2F and EV2C). PIAS4 depletion also severely inhibited poly-SUMOylation of crosslinked SUMO-M.HpaII, although in this case residual SUMOylation of the DPC was observed, suggesting that in the absence of PIAS4 another SUMO E3 ligase may contribute to extending some M.HpaII SUMO modifications (Fig 2G). Importantly, M.HpaII DPC poly-SUMOylation was restored by supplementing extracts with recombinant WT PIAS4 but not a mutant lacking its dual SIMs (Kaur *et al*, 2017), which showed defective accumulation on damaged chromatin (Figs 2F and G, and EV2D–G), consistent with the SUMO-dependent enrichment of PIAS4 detected by CHROMASS. PIAS4 was also responsible for PARP1 SUMOylation both in the presence or absence of DNA damage (Fig EV2H and I). Furthermore, purified PIAS4 stimulated poly-SUMOylation of M.HpaII DPCs *in vitro* (Fig 2H). Collectively, these data show that DPC SUMOylation can occur in the complete absence of DNA replication and suggest a primary role of PIAS4 in this response in *Xenopus* egg extracts.

In cells, consistent with our observations in *Xenopus* egg extracts, PIAS4 was recruited to DNMT1 DPC sites in a SUMOylation-dependent manner and displayed increased binding to DNMT1 upon DPC formation (Figs 2I and J, and EV2J), suggesting its engagement in promoting SUMO modification of these DPCs. However, in contrast to its major role observed in *Xenopus* egg extracts, depletion of PIAS4 had no overt impact on DNMT1 DPC SUMOylation (Figs 2K and EV2K and L). In fact, individual depletion of established SUMO E3 ligases did not impair overall DNMT1 DPC SUMOylation (Fig 2K), suggesting a possible redundancy between PIAS4 and other SUMO E3s in driving this response, in line with its strong magnitude. Supporting this notion, SUMO-dependent

recruitment of PIAS4 to DNMT1 DPCs occurred normally in cells devoid of PIAS4 catalytic activity (Fig EV2M and N), suggesting the involvement of one or more other SUMO E3s in promoting PIAS4 accumulation at DPC sites. The differential requirement of PIAS4 for DPC SUMOylation in cells and *Xenopus* egg extracts could reflect the distinct chromatin environments harbouring the DPCs in the two systems and/or the absence in egg extracts of active transcription, which may provide an additional DPC-sensing mechanism in cells.

## DPC SUMOylation neither inhibits nor stimulates replication-coupled DPC repair

In contrast to DNA replication-independent DPC SUMOylation, our previous work showed that during DNA replication DPCs are rapidly ubiquitylated by TRAIP and subsequently degraded by SPRTN and the proteasome (Larsen *et al*, 2019). We therefore utilized the notion that SUMOylated DPCs are stable in NPE to analyse the impact of DPC SUMOylation on replication-coupled DPC repair. To this end, a plasmid in which M.HpaII is crosslinked to the leading strands (p2xDPC$^{Leads}$) was first replicated in egg extract in the presence or absence of ubiquitin or SUMO E1 inhibitors. During replication of p2xDPC$^{Leads}$, converging forks transiently stall at the DPC, after which daughter plasmid molecules are resolved (Larsen *et al*, 2019) (Fig 3A). The daughter molecules containing the DPC initially migrate as an open circular (OC) species (Fig 3B, lane 2 (red arrowhead)) and are then gradually converted to a supercoiled (SC) repair product through proteolysis of the DPC and translesion DNA synthesis (TLS) across the resulting peptide adduct (Fig 3B, lanes 1–5). As expected, inhibition of ubiquitylation suppressed replication-coupled

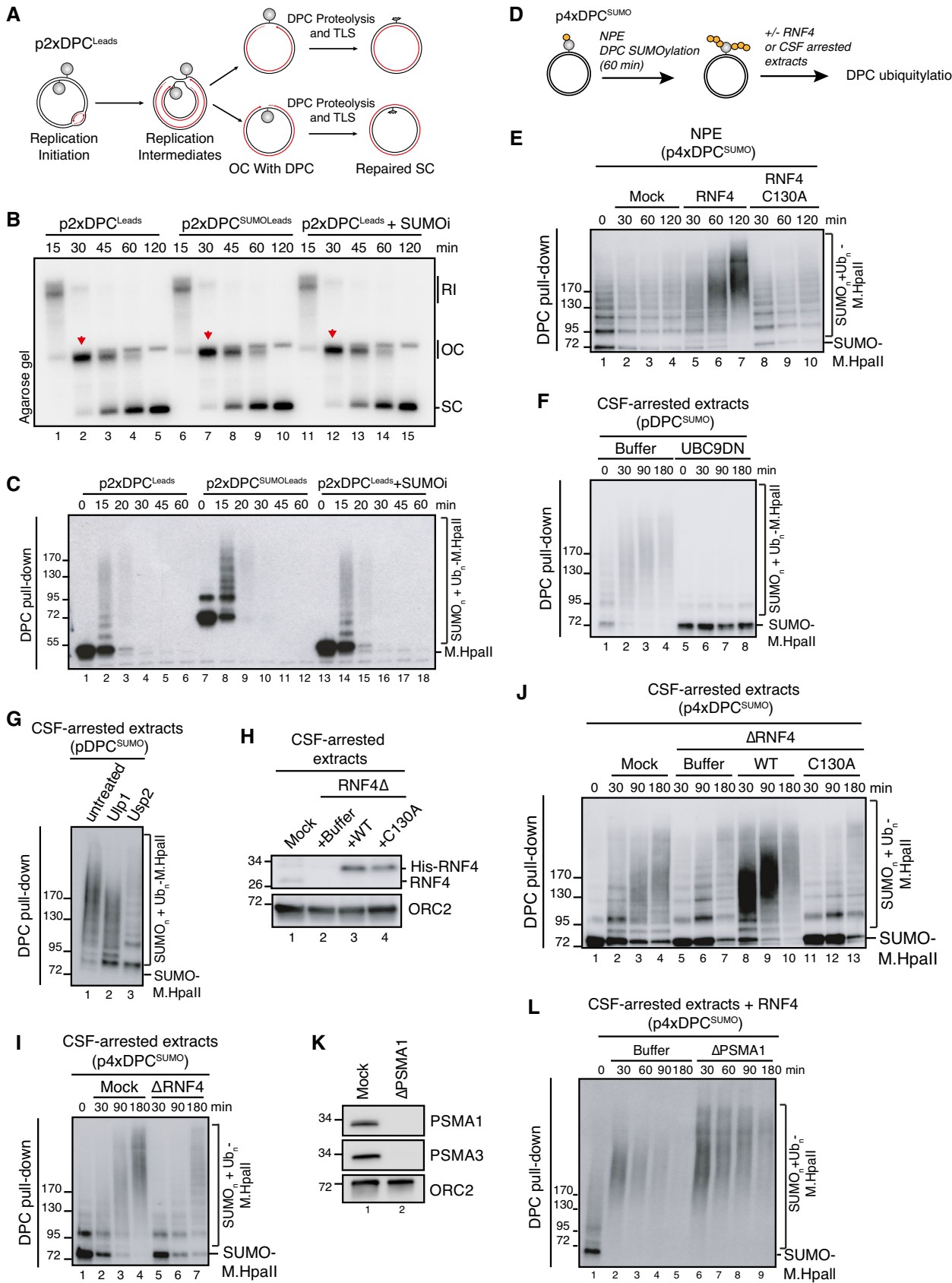

Figure 3.

**Figure 3.  The SUMO-RNF4 DPC resolution pathway is independent of DNA replication and neither stimulates nor inhibits replication-coupled DPC repair.**

A    Scheme illustrating replication in egg extracts of p2xDPC$^{Leads}$, a plasmid containing two M.HpaII crosslinked on opposite strands. Under these conditions, the vast majority of replication forks encounter the DPC on their leading strand template (Larsen *et al*, 2019).

B    p2xDPC$^{Leads}$ or p2xDPC$^{SUMOLeads}$ were replicated in egg extracts in the presence [α-$^{32}$P]dATP. Where indicated, 50 μM of SUMOi was added to the extracts. Reaction samples were analysed by native agarose gel electrophoresis. RI, replication intermediates; OC, open circular; SC, supercoiled. Red arrowheads indicate OC molecules that have not yet undergone repair.

C    Samples in (B) were recovered by DPC pull-down and immunoblotted for crosslinked M.HpaII.

D    Scheme illustrating sequential SUMOylation and ubiquitylation of DPC-containing plasmids in non-replicating egg extracts. First, p4xDPC$^{SUMO}$ is incubated in NPE for 60-90 min to achieve poly-SUMOylation of the DPCs. NPE is then supplemented with recombinant RNF4 to a final concentration of 7 ng/μl (E) or an equal volume of CSF-arrested whole egg extract (F, G) to trigger DPC ubiquitylation.

E    p4xDPC$^{SUMO}$ was incubated in NPE for 60 min and supplemented with buffer or recombinant RNF4 (WT or a catalytically inactive C130A mutant). At the indicated time points following RNF4 addition, the DPC plasmid was recovered by DPC pull-down and immunoblotted against M.HpaII.

F    pDPC$^{SUMO}$ was subjected to sequential extract addition as depicted in (D). Recombinant UBC9DN was added to NPE where indicated to block *de novo* SUMOylation. At the indicated time points following addition of CSF-arrested egg extract, the plasmid was recovered via DPC pull-down and immunoblotted against M.HpaII.

G    pDPC$^{SUMO}$ was subjected to sequential extract addition as depicted in (D). Ninety min after addition of CSF-arrested extract, samples were recovered via DPC pull-down and treated with the SUMO protease Ulp1 or the ubiquitin protease Usp2 as indicated. Samples were then immunoblotted against M.HpaII.

H    CSF-arrested extracts were either mock- or RNF4-depleted and recombinant His-RNF4 was supplemented to RNF4-depleted extracts to a final concentration of 7 ng/μl where indicated. Protein samples were immunoblotted with the indicated antibodies.

I     p4xDPC$^{SUMO}$ was polySUMOylated in NPE, recovered via DPC pull-down and incubated in fresh CSF-arrested extract that was either mock- or RNF4-depleted. At indicated time points following CSF extract addition, the plasmid was recovered and immunoblotted against M.HpaII.

J     As in (I), but using RNF4-depleted CSF-arrested extracts reconstituted with recombinant RNF4 WT or C130A from (H).

K    CSF-arrested extracts were either mock- or PSMA1-depleted. Protein samples were immunoblotted with the indicated antibodies.

L     p4xDPC$^{SUMO}$ was subjected to sequential addition of NPE and CSF-arrested extract as in (I). CSF-arrested extracts from (K) were supplemented with recombinant RNF4, and the DPC plasmid was recovered by DPC pull-down at the indicated time points and immunoblotted against M.HpaII.

DPC repair, evidenced by the accumulation of OC molecules, while addition of SUMOi had no effect (Fig 3B, lanes 11–15; Fig EV3A). Consistently, M.HpaII DPCs were ubiquitylated and degraded normally in the presence of SUMOi (Fig 3C, lanes 13–18; Fig EV3B). Likewise, SUMO-fused M.HpaII, which induced some poly-SUMOylation during the licensing reaction (Fig 3C, lane 7), did not impair DPC ubiquitylation and repair during DNA replication (Fig 3B, lanes 6–10; Fig 3C, lanes 7–12), nor in the absence of a replication fork when the DPC was placed on ssDNA (Fig EV3C and D). Interestingly, PIAS4-dependent DPC SUMOylation *in vitro* was not restricted to DPCs on dsDNA but also occurred on DPCs on ssDNA (Fig 2H), suggesting that in contrast to replication-dependent mechanisms that target DPCs on ssDNA, the SUMO pathway can sense DPCs residing in different DNA contexts. We conclude that DPC SUMOylation is neither inhibitory nor stimulatory towards replication-coupled DPC repair. Instead, it likely provides an alternative and independent modification route when DPC repair cannot be accomplished via DNA replication-associated mechanisms.

## RNF4 targets SUMO-modified DPCs in a DNA replication-independent manner

Unlike DNMT1 DPCs in human cells, DPC SUMOylation in NPE did not trigger RNF4-mediated ubiquitylation and degradation. By generating an antibody against *Xenopus* RNF4, we realized that it is only present at low nanomolar concentrations in egg extracts (Fig EV3E), potentially explaining why SUMOylated DPCs were not further modified by ubiquitylation. Importantly, by supplementing NPE with recombinant RNF4 we could recapitulate DPC ubiquitylation in the complete absence of DNA replication, and this effect was dependent on RNF4 catalytic activity (Figs 3D and E, and EV3F). A similar effect was observed when supplementing NPE with whole egg CSF-arrested extracts, which contain slightly higher concentrations of RNF4 (Fig EV3E). In this setting, the mitosis-like extract induced DPC modifications that contained both ubiquitin and SUMO moieties

and were fully suppressed when *de novo* SUMOylation was inhibited by addition of a dominant-negative SUMO E2 enzyme (UBC9DN) (Fig 3F and G). Likewise, DPC ubiquitylation could be observed when SUMOylated DPCs were isolated following incubation in NPE and transferred into fresh CSF-arrested extracts (Fig 3I, lane 1-3). This ubiquitylation was impaired by depletion of RNF4 and restored by adding back recombinant WT but not catalytically inactive RNF4 to the extracts (Fig 3H–J). Correspondingly, while CSF-arrested extracts were less efficient than NPE in promoting M.HpaII DPC SUMOylation and had lower PIAS4 concentration, DPC SUMOylation and ubiquitylation in this extract could be strongly enhanced by addition of recombinant PIAS4 (Fig EV3G–I). Addition of WT RNF4 markedly accelerated ubiquitylation of SUMOylated DPCs in both CSF-arrested extracts (Fig 3J, lanes 8–10) and NPE (Fig EV3J and K). Under these conditions the DPCs underwent degradation, which was partially inhibited upon proteasome depletion (Fig 3K and L, and EV3L and M) or inhibition (Fig EV3K, lanes 11–13), as observed in cells. Moreover, addition of a pan-CDK inhibitor to CSF-arrested extracts inhibited RNF4-dependent DPC ubiquitylation (Fig EV3N and O), consistent with a previous report indicating that RNF4 E3 ligase activity is stimulated by CDK activity (Luo *et al*, 2015). We conclude from these findings that SUMOylated DPCs are directly targeted by RNF4 STUbL activity in both human cells and *Xenopus* egg extracts and that this process occurs in the complete absence of DNA replication. Moreover, RNF4-dependent DPC ubiquitylation may be potentiated in a post-replicative manner by the progressive cell cycle-dependent increase in cyclin-CDK activity.

## Unresolved DPCs in duplex DNA do not elicit an effective DNA damage checkpoint

The mechanistic insights into SUMO-driven pathways for DNA replication-independent resolution of DNMT1 DPCs described above provided an opportunity to address if and how unresolved DPCs on duplex DNA undermine genome stability and cellular fitness. To

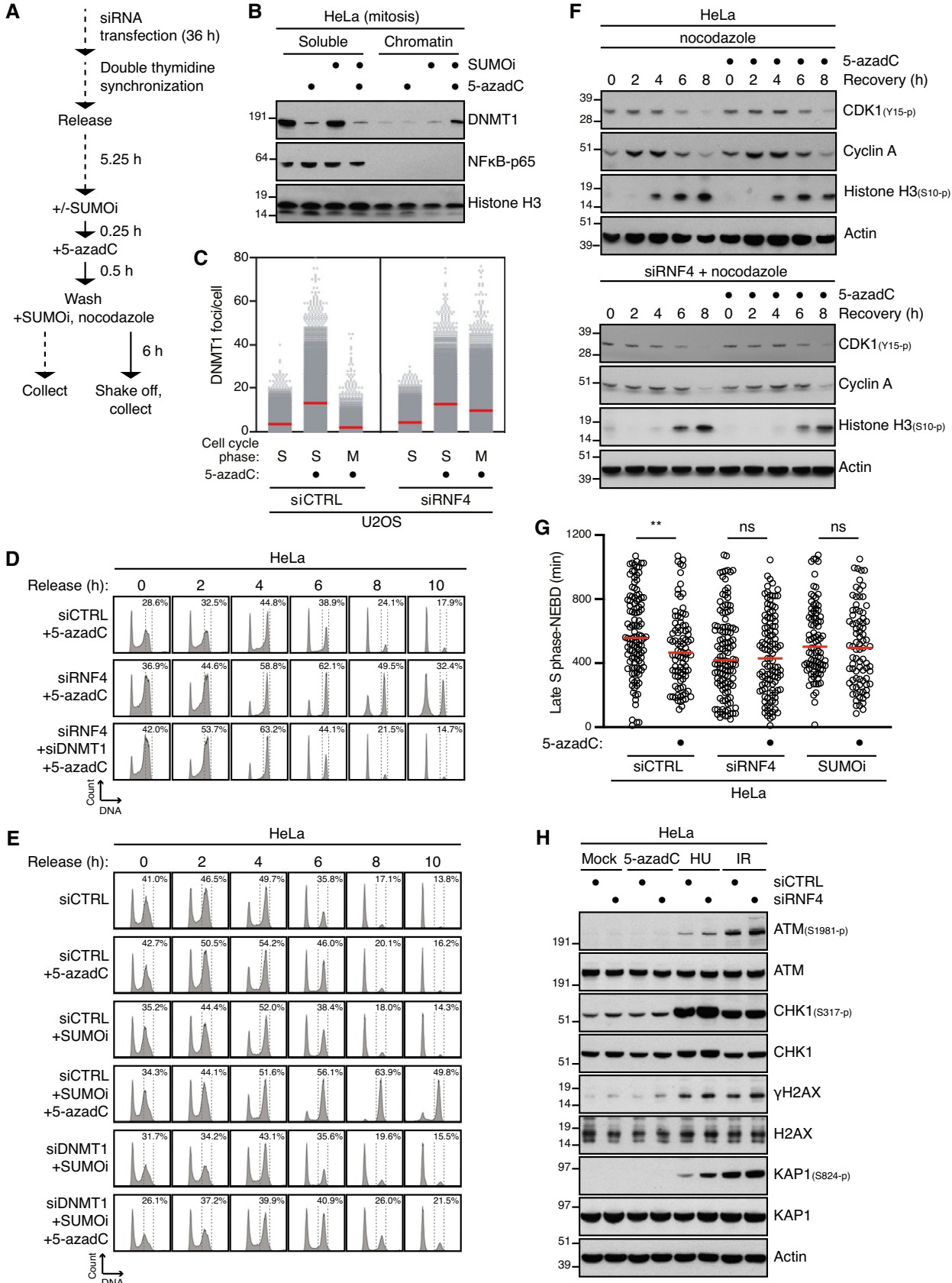

**Figure 4.**

**Figure 4. DPCs on duplex DNA do not activate the DNA damage checkpoint.**

A   Outline of experimental set up to monitor DNMT1 DPC levels in S phase and mitotic cells.
B   HeLa cells released from double thymidine block in early S phase were mock-treated or pulse-labelled for 30 min with 5-azadC in the presence of SUMOi in late S phase. Following 5-azadC removal, cells were incubated with SUMOi and nocodazole, and mitotic cells were collected by shake-off, as outlined in (A). Soluble and chromatin-enriched fractions were immunoblotted with indicated antibodies.
C   U2OS cells transfected with indicated siRNAs and then treated as in (B) were collected in late S phase or mitosis (M), subjected to stringent pre-extraction and immunostained with DNMT1 antibody. DNMT1 foci formation was quantified by QIBC analysis (red bars, mean; > 7,400 cells analysed per condition). Data are representative of three independent experiments.
D   HeLa cells transfected with indicated siRNAs and synchronized in early S phase by double thymidine block were pulse-labelled with 5-azadC in late S phase as outlined in (A). Cells were then collected at the indicated times after 5-azadC withdrawal and analysed by flow cytometry. Data are representative of three independent experiments. Proportion of cells with G2/M DNA content is indicated.
E   HeLa cells transfected with indicated siRNAs were treated with 5-azadC and/or SUMOi in late S phase as outlined in (A). Cells were then collected at the indicated times after 5-azadC withdrawal and analysed by flow cytometry. Data are representative of three independent experiments. Proportion of cells with G2/M DNA content is indicated.
F   HeLa cells transfected with indicated siRNAs were pulse-labelled or not with 5-azadC for 30 min in late S phase according to the experimental setup in (A). Following 5-azadC withdrawal cells were incubated with nocodazole, collected at the indicated times and immunoblotted with indicated antibodies.
G   HeLa cells transfected or not with indicated siRNAs were synchronized in early S phase by double thymidine block, released and pulse-labelled 5,5 h later with 5-azadC for 30 min in the presence or absence of SUMOi. Following 5-azadC removal, cells were subjected to live-cell imaging analysis, and the duration from late S phase to mitotic entry (nuclear envelope breakdown (NEBD)) was quantified (red bars, median; at least 83 cells, pooled from three independent experiments, were analysed per condition; **$P$ < 0.01, ns: not significant, Mann–Whitney test).
H   Immunoblot analysis of siRNA-treated HeLa cells that were exposed to indicated genotoxic agents and collected 1 h later.

explore this, we pulse-labelled cells in late S phase with 5-azadC and subsequently monitored the impact of RNF4 knockdown or SUMO inhibition on DNMT1 DPC removal and cell cycle progression (Fig 4A). As expected, 5-azadC-induced DPCs generated in late S phase were largely resolved by the time control cells entered mitosis (Fig 4B and C), consistent with their robust capacity for DNMT1 DPC resolution (Fig 1B). By contrast, both RNF4 depletion and SUMOi treatment led to a persistence of DNMT1 DPCs in mitotic cells (Fig 4B and C). Strikingly, whereas neither 5-azadC treatment nor RNF4 depletion alone delayed cell cycle progression following synchronization in early S phase, combined RNF4 knockdown and 5-azadC exposure led to a prominent accumulation of cells in G2/M phase (Figs 4D and EV4A). Combining SUMOi and 5-azadC treatment caused an even stronger accrual of cells in G2/M phase (Fig 4E), in agreement with the more severe impact of SUMOi on DNMT1 DPC resolution relative to RNF4 knockdown. Importantly, knockdown of DNMT1 rescued the accumulation of G2/M phase cells resulting from inhibition of SUMOylation or RNF4 depletion in 5-azadC-treated cells (Fig 4D and E), demonstrating that it was a specific consequence of DNMT1 DPC formation. Several lines of evidence from HeLa and non-transformed RPE-1 cells suggested that impairment of DNMT1 DPC resolution leads to accumulation of cells in mitosis rather than in G2 phase. First, exposing RNF4-depleted or SUMOi-treated cells released from early S phase arrest to 5-azadC had no overt impact on the kinetics with which the mitotic marker Histone H3-pSer10 accumulated (Figs 4F and EV4B–D). Second, live-cell imaging experiments showed that the timing of mitotic entry, as assessed by nuclear envelope breakdown, of cells pre-synchronized in S phase was not significantly altered by combined 5-azadC and SUMOi or RNF4 siRNA treatment (Fig 4G). Finally, unlike genotoxic insults such as DNA breakage and replication stress generated by exposure to ionizing radiation (IR) and hydroxyurea (HU), respectively, 5-azadC-induced DNMT1 DPCs did not trigger detectable DNA damage checkpoint signalling in HeLa and RPE-1 cells, regardless of whether DPC resolution was impaired by RNF4 depletion (Figs 4H and EV4E). Likewise, whereas it potently

induces DPCs and an accompanying strong chromatin SUMOylation response (Borgermann et al, 2019), formaldehyde only weakly activated canonical DNA damage signalling, possibly reflecting its ability to also generate other types of DNA damage (Fig EV4F). Together, these findings suggest that human cells do not elicit an effective DNA damage checkpoint response to DPCs in otherwise undamaged duplex DNA and thus enter mitosis with normal kinetics even in the face of a high load of these lesions, in contrast to most other types of DNA damage.

## Unresolved DPCs undermine faithful mitotic chromosome segregation

Despite the apparent lack of a G2 checkpoint for sensing DPCs on duplex DNA, preventing the timely resolution of these lesions nevertheless caused prominent accumulation of cells with G2/M DNA content (Fig 4D and E). This suggested that unresolved DPCs might interfere with proper progression through mitosis. To test this, we used live-cell imaging to monitor how unresolved DPCs affect chromosome segregation and cell division. While treatment with 5-azadC, SUMOi or RNF4 siRNA alone had no significant impact on the kinetics of mitotic progression, we observed a substantial delay to anaphase onset following mitotic entry (marked by nuclear envelope breakdown) in siRNF4- or SUMOi-treated cells exposed to 5-azadC, consistent with their accumulation in G2/M phase (Fig 5A). Detailed inspection of mitotic progression revealed that a large proportion of cells with unresolved DNMT1 DPCs showed a marked delay in completing chromosome alignment at the metaphase plate, giving rise to defective chromosome segregation (Fig 5B and C). Moreover, cells with high loads of 5-azadC-induced DNMT1 DPCs due to RNF4 knockdown frequently displayed abortive cytokinesis leading to nuclear abnormalities (Fig 5D). Suppression of the spindle assembly checkpoint (SAC) using an MPS1 kinase inhibitor (MPS1i) accelerated mitotic exit in RNF4-depleted cells harbouring unresolved DNMT1 DPCs and enhanced aneuploidy (Figs 5E and EV5A), suggesting that unlike interphase DNA

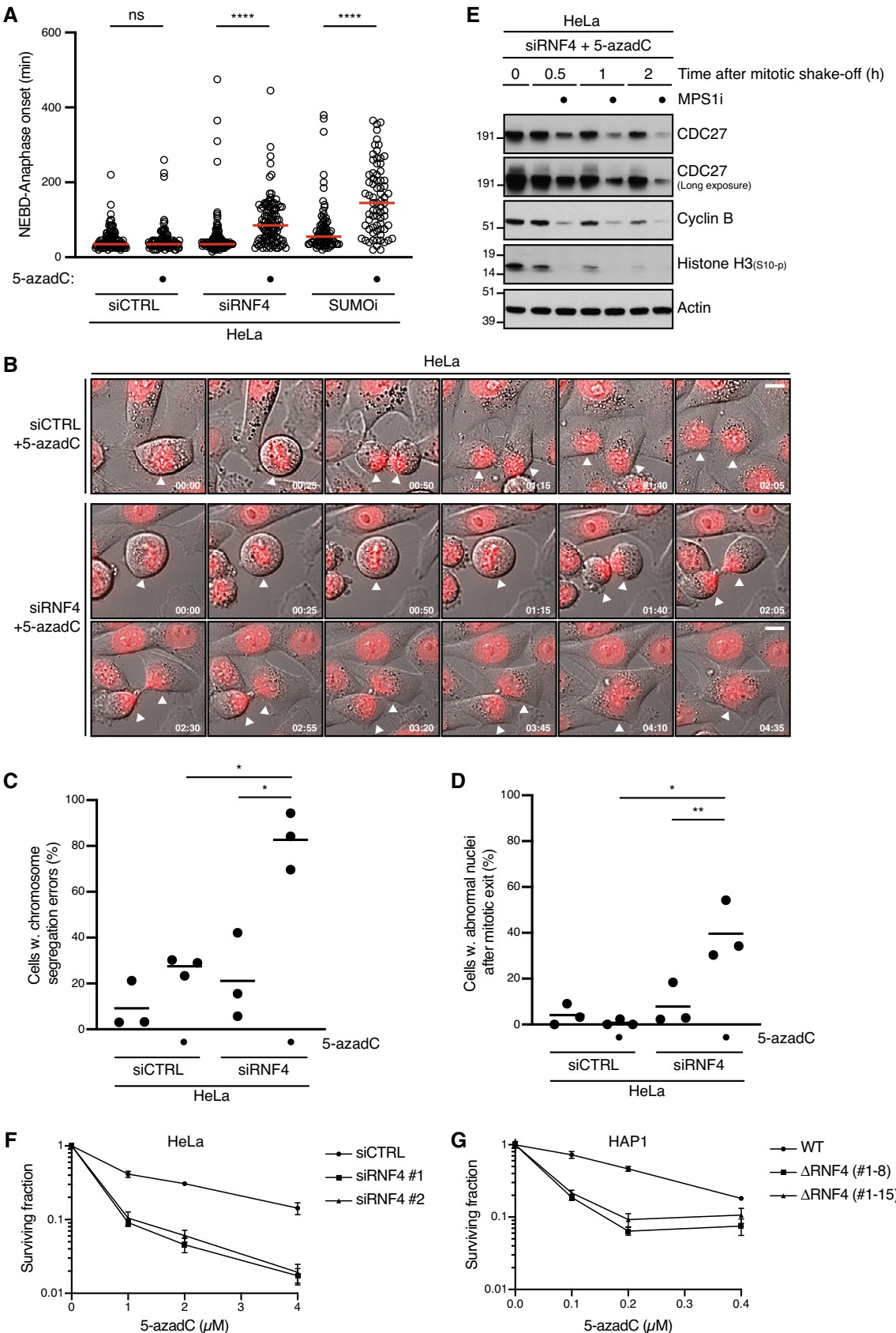

**Figure 5.**

**Figure 5. Unresolved DPCs undermine faithful mitotic chromosome segregation and cellular fitness.**

A   HeLa cells transfected or not with indicated siRNAs were synchronized in early S phase by double thymidine block, released and pulse-labelled 5,5 h later with 5-azadC for 30 min in the presence or absence of SUMOi. Following 5-azadC removal, cells were subjected to live-cell imaging analysis, and the duration of mitosis (nuclear envelope breakdown (NEBD) to anaphase onset) was quantified (red bars, median; at least 72 cells, pooled from three independent experiments, were analysed per condition; ****$P$ < 0.0001, ns: not significant, Mann–Whitney test).

B   Representative time-lapse sequences of mitotic progression in cells in (A). NEBD corresponds to $t$ = 0. White arrowheads indicate cells undergoing division. Scale bar, 10 μm.

C, D   Quantification of mitotic defects in cells in (A) (mean; $n$ = 3 independent experiments; > 97 cells quantified per condition; *$P$ < 0.05, **$P$ < 0.01, paired $t$-test).

E   HeLa cells transfected with RNF4 siRNA were synchronized in early S phase by double thymidine block. Six hours after release from the block, cells were pulse-labelled with 5-azadC for 30 min. Mitotic cells were isolated by shake-off 8 h later, washed and incubated or not with MPS1 inhibitor (MPS1i) and collected at the indicated times. Cells were then processed for immunoblotting with indicated antibodies.

F   Clonogenic survival of 5-azadC-treated HeLa cells transfected with indicated siRNAs (mean ± SEM; $n$ = 2 independent experiments).

G   Clonogenic survival of 5-azadC-treated WT HAP1 cells and derivative cell lines expressing a truncated form of RNF4 lacking the RING domain (ΔRNF4; Fig EV1H) (mean ± SEM; $n$ = 2 independent experiments).

damage checkpoints DPCs on duplex DNA trigger activation of the SAC. Importantly, formaldehyde-induced DPCs similarly prolonged mitosis and led to a markedly elevated rate of defective chromosome segregation when their resolution was blocked by SUMOi (Fig EV5B and C), suggesting this is a general effect of unrepaired DPCs in otherwise undamaged duplex DNA. Finally, consistent with the extensive chromosome segregation defects and resulting genomic instability incurred by cells undergoing mitosis in the presence of unresolved DNMT1 DPCs, loss of RNF4 functionality by independent siRNAs or CRISPR/Cas9-mediated truncation hypersensitized cells to 5-azadC (Figs 5F and G, and EV1C and H). Cells lacking

RNF4 also displayed increased sensitivity to formaldehyde (Fig EV5D). Collectively, these data establish and underscore a critical role of post-replicative, SUMO-driven DPC resolution mechanisms in the maintenance of chromosome stability and cellular fitness.

# Discussion

Our recent work established an essential role of protein SUMOylation in promoting resolution of DPCs, impinging directly on the

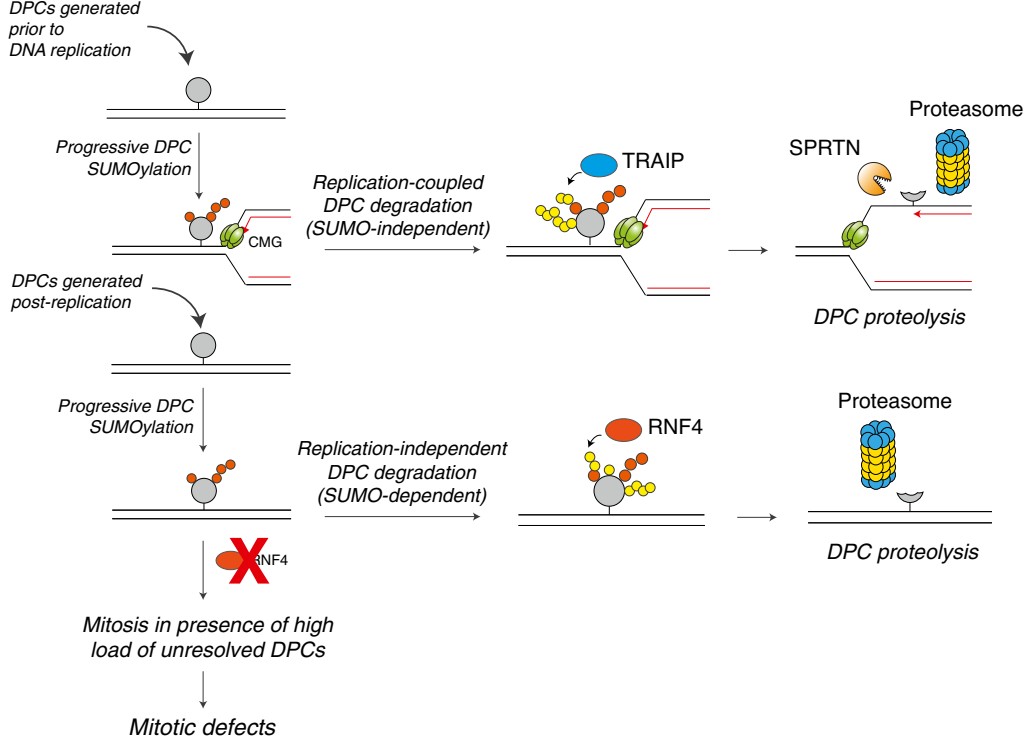

**Figure 6. Mechanism of SUMO-mediated DNA replication-independent DPC repair.**

Scheme illustrating independent mechanisms of replication-coupled and SUMO-mediated replication-independent DPC repair available to cells. In the absence of RNF4-mediated DPC resolution, cells containing high loads of DPCs generated after DNA replication fail to elicit a DNA damage checkpoint response and consequently enter mitosis with unresolved DPCs, leading to chromosome segregation defects.

crosslinked protein (Borgermann *et al*, 2019). However, mechanistically how SUMOylation of DPCs on duplex DNA drives their removal and its significance for genome stability maintenance remained unclear. In addition, while available evidence suggests that cells possess both DNA replication-coupled and -independent mechanisms for clearing DPCs, the interrelationship between these processes has not been directly studied. In the present study, we addressed these important questions using both human cells and *Xenopus* egg extracts. We demonstrate that the STUbL activity of RNF4 towards SUMOylated DPCs is fully uncoupled from DNA replication and provides a major salvage route that is critical for removing DPCs prior to mitosis and thereby ensure proper cell division and cellular fitness (Fig 6).

**Replication-coupled and replication-independent DPC repair pathways**

While both replication-coupled and replication-independent resolution pathways can promote DPC proteolysis via the proteasome, the former proceeds via direct DPC ubiquitylation by dedicated replication fork-associated or ssDNA targeted ubiquitin ligases including TRAIP and RFWD3 (Larsen *et al*, 2019; Gallina *et al*, 2021), whereas outside the context of DNA replication this is mediated indirectly by the coupling of SUMOylation and RNF4 STUbL activity. Importantly, we provide direct evidence that DPC SUMOylation neither interferes with nor stimulates replication-coupled DPC resolution, indicating that these modes of DPC processing represent fully independent repair pathways. Indeed, both DPC ubiquitylation and degradation by SPRTN and/or the proteasome were unaffected when the DPC was SUMOylated prior to DNA replication or if DNA replication occurred in the absence of *de novo* SUMOylation. Thus, SUMOylated DPCs, which are likely generated throughout the cell cycle and could be encountered by the replication machinery, are unlikely to compete with rapid SPRTN- and ubiquitin/proteasome-driven DPC proteolysis at the replication fork or ssDNA gaps (Fig 6). Instead, our new findings suggest and provide a mechanistic rationale for a critical role of SUMO-driven DPC resolution in complementing replication-coupled DPC repair by either clearing the lesions prior to DNA replication or removing DPCs generated after the passage of replication forks, the latter process being essential for subsequent faithful chromosomal segregation and cell division (Fig 6). In agreement with our data, a recent study showed that topoisomerase-type crosslinks, which are flanked by DNA breaks, can also be targeted for proteasomal degradation by RNF4 (Sun *et al*, 2020). Likewise, previous studies described an essential role of STUbLs in counteracting topoisomerase DPCs in yeast (Heideker *et al*, 2011; Steinacher *et al*, 2013; Wei *et al*, 2017; Sun *et al*, 2020). SUMO-driven ubiquitylation and removal of DPCs thus emerges as a conserved pathway for eukaryotic cells to resolve these toxic lesions, and it seems likely that this mechanism can target a wide range of DPCs encountered in the genome. Of note, repair of TOP2 DPCs can also occur in the absence of DPC proteolysis via the ZATT-TDP2 axis, which may represent the default option for resolving these lesions (Schellenberg *et al*, 2017). Thus, RNF4-mediated DPC destruction might provide an alternative or back-up pathway when target-specific DPC repair processes are not available due to the nature of the crosslink or cell cycle stage.

**The SUMO-RNF4 DPC resolution pathway is essential for faithful cell division**

Surprisingly, although DNMT1 DPCs generated behind the replication fork are efficiently sensed by cells as evidenced by their robust crosslinking-dependent SUMOylation and subsequent ubiquitylation, we found that these DPCs do not trigger detectable levels of conventional DNA damage signalling, suggesting that cells do not mount an effective interphase checkpoint response to DPCs in otherwise undamaged dsDNA and consequently fail to prevent cells harbouring unresolved DPCs from entering mitosis. This in turn has a severe impact on chromosome stability, as demonstrated by our finding that a large proportion of cells in which DNMT1 DPC repair is suppressed by RNF4 knockdown display defective chromosome segregation leading to aneuploidy. Preventing SUMOylation-dependent removal of formaldehyde-induced DPCs likewise undermines faithful mitotic chromosome segregation, suggesting this is a general feature of unresolved DPCs that are not accompanied by DNA breakage. While it seems likely that this inability to trigger activation of the DNA damage checkpoint may apply selectively to DPCs in uninterrupted duplex DNA such as those generated by 5-azadC or reactive aldehydes, the potential of these lesions for undermining accurate chromosome segregation clearly underscores the importance of efficient SUMO-driven post-replicative DPC resolution mechanisms for maintaining chromosomal integrity, considering also that the formaldehyde dose used in our experiments is close to its estimated intracellular concentration in human cells (approx. 400 μM; European Food Safety Authority, 2014). One potential mechanism for facilitating efficient post-replicative resolution of SUMOylated DPCs could entail cyclin-CDK-driven stimulation of their RNF4-mediated ubiquitylation, in keeping with previous work showing that RNF4 E3 ubiquitin ligase activity is regulated by stimulatory phosphorylation by CDKs (Luo *et al*, 2015). Indeed, we found that mitosis-like egg extract potentiated RNF4-mediated DPC ubiquitylation in a CDK-dependent manner. Thus, the cell cycle-dependent increase in cyclin-CDK activity may progressively stimulate RNF4-mediated ubiquitylation of SUMO-modified DPCs, ensuring their rapid removal during late stages of the cell cycle. We note in this context that levels of reactive oxygen species (ROS), which fuel the generation of aldehydes that are strong drivers of DPC formation, have been shown to fluctuate during the cell cycle and peak in G2 phase (Singh *et al*, 2013; Patterson *et al*, 2019). Based on our findings, we propose that the SUMO-RNF4 axis provides a crucial salvage pathway for efficient clearance of DPCs formed after DNA replication but prior to mitotic entry, thus protecting against the deleterious impact of DPCs on faithful chromosome segregation (Fig 6). Precisely how unresolved DPCs undermine the integrity of mitosis remains to be established, but it is possible that a high load of DPCs could compromise the initial steps of chromosome condensation.

**Is SUMOylation a global modulator of protein residence time on DNA?**

A body of previous work has shown that in addition to DPCs, many non-covalently bound proteins are specifically modified by SUMOylation when associated with chromatin to promote their displacement from this compartment, at least in some cases mediated by STUbL activity (Galanty *et al*, 2012; Luo *et al*, 2012; Gibbs-

Seymour *et al*, 2015; Guérillon *et al*, 2020). Based on these observations, it is tempting to speculate that SUMOylation may provide a general mechanism for limiting the interaction of proteins with DNA or chromatin, with the magnitude of SUMOylation correlating with the residence time of the bound protein and/or its affinity for DNA or chromatin. In such a scenario, DPCs represent an extreme case given their covalent trapping on DNA, providing an attractive rationale for the remarkably potent SUMOylation response targeting 5-azadC- and formaldehyde-induced DPCs in cells (Borgermann *et al*, 2019). In further agreement with this model, we found that the non-covalent trapping of PARP1 on DNA by PARP inhibitors also triggers its SUMO-dependent modification. Likewise, it is well established that many transcription factors and chromatin remodellers are targeted by SUMOylation. For instance, several studies have shown that the SUMOylation of some transcription factors reduces their affinity for or occupancy time on DNA (Rosonina *et al*, 2012; Tempé *et al*, 2014; Akhter & Rosonina, 2016). Thus, SUMOylation appears to be a mark placed on a multitude of diverse DNA-interacting proteins to curb their association with DNA. In some instances, SUMOylation of weak DNA interactors might be sufficient to lower their affinity for chromatin, whereas for tightly bound proteins or DPCs, SUMOylation may require subsequent ubiquitylation and degradation by the proteasome. The recognized substrate promiscuity of protein SUMOylation and its link to the ubiquitin-proteasome system via STUbLs could render this modification well suited as a non-specific modifier of multitudinous DPCs and other non-covalently bound proteins whose chromatin occupancy must be constrained to maintain the integrity and functionality of the genome. Although we found that PIAS4 is clearly the main mediator of M.HpaII DPC SUMOylation in *Xenopus* egg extracts, catalyses DPC SUMOylation *in vitro* and is recruited to DNMT1 DPC sites in cells, it appears to act redundantly with one or more other SUMO E3s in driving the dramatic cellular DNMT1 SUMOylation response accompanying its crosslinking to DNA. Consistent with an ability of PIAS4 to SUMOylate different DPCs, it was recently shown to SUMOylate TOP1 and TOP2 DPCs in cells, although also in this setting its absence only had a partial impact on TOP1/TOP2 DPC SUMOylation (Sun *et al*, 2020).

Together with the recent discoveries of several DPC removal enzymes, our work highlights the versatile toolbox available to cells to resolve these toxic lesions at different stages of the cell cycle. In some cases, these DPC-processing enzymes are uniquely tailored for specific types of crosslinks, as in the case of TOP1/2 adduct removal by TDP1/2. When specialized DPC resolution pathways cannot be utilized, cells instead rely on more generic complementary DNA replication-coupled and SUMO-driven replication-independent repair pathways to non-discriminately remove the large diversity of DPCs and thereby mitigate the threat they pose to genome integrity.

# Materials and Methods

## Cell culture

Human HeLa, U2OS and RPE-1 cells were obtained from ATCC. HeLa and U2OS cells were propagated in Dulbecco's modified Eagle's medium (DMEM) and RPE-1 cells were grown in mix of DMEM and F-12 medium supplemented with 10% (v/v) foetal bovine serum and 1% penicillin–streptomycin. Cells were cultured in humidified incubators at 37°C with 5% $CO_2$ and were regularly tested negative for mycoplasma infection. The cell lines were not authenticated. HAP1 cells were cultured in IMDM-medium (Gibco) supplemented with 10% heat-inactivated foetal calf serum (FCS; Thermo Fisher Scientific) and penicillin–streptomycin–glutamine solution (Gibco). All HAP1 cell lines were monitored for ploidy and authenticated by genotyping. HAP1 cells were co-transfected with a blasticidin resistance vector and the CRISPR/Cas9 plasmid pX330 containing a RNF4 targeting sgRNA (5′-CTGCATGGACGGATAC TCAG-3′). Cells were selected in 20 µg/ml blasticidin for 2 days and clonally expanded to generate isogenic cell lines. RNF4 KO cell lines were validated by DNA sequencing. HeLa cells stably expressing GFP-DNMT1 (HeLa/GFP-DNMT1) were described previously (Borgermann *et al*, 2019). Plasmids encoding Myc-PIAS4 WT or a catalytically inactive C342A mutant (CI) were generated by inserting the corresponding cDNAs into pcDNA3 (Invitrogen) containing an N-terminal Myc-tag. Plasmid DNA transfections were performed using FuGENE 6 (Promega), Novagen GeneJuice (Merck) or Lipofectamine 2000 (Invitrogen), according to the manufacturers' protocols. Cell cycle synchronizations were performed as described (Ma & Poon, 2011). Briefly, HeLa, U2OS or RPE-1 cells were synchronized at the G1/S transition by incubating cells with 2 mM thymidine for 13–15 h, followed by 9-h release with fresh medium and subsequent thymidine incubation for an additional 13–15 h. HAP1 cells were synchronized by single treatment with thymidine for 16 h. Mitotic cells were obtained by treatment with nocodazole followed by mechanical shake-off.

Unless otherwise indicated, the following doses of drugs and genotoxic agents were used: 5-aza-2′-deoxycytidine (5-azadC; 10 µM, Sigma-Aldrich), formaldehyde (500 µM, Thermo Scientific Pierce), aphidicolin (APH; 10 µM, Sigma-Aldrich), hydroxyurea (HU; 2 mM, Sigma-Aldrich), MG132 (20 µM, Sigma-Aldrich) ML-792 (SUMOi; 2 µM, synthesized by MedKoo Biosciences), MLN-7243 (UBi; 5 µM, Active Biochem), nocodazole (0.04 µg/ml, Sigma-Aldrich), reversine (MPS1i; 0.5 µM, Cayman Chemicals), thymidine (2 mM, Sigma-Aldrich) and ionizing radiation (IR; 10 Gy).

## siRNAs

siRNA transfections were performed using Lipofectamine RNAiMAX (Invitrogen) according to the manufacturer's instructions. All siRNAs were used at a final concentration of 20 nM unless otherwise indicated. The following siRNA oligonucleotides, whose knockdown efficiencies were validated where indicated, were used: non-targeting control (CTRL): 5′-GGGAUACCUAGACGUUCUA-3′; MMS21: 5′-CUCUGGUAUGGACACAGCUTT-3′ (Potts & Yu, 2005); RanBP2: 5′-GGACAGUGGGAUUGUAGUGTT-3′ (Joseph *et al*, 2004); PIAS1: 5′-UAUUAAUGUAGCUUGUGUCUACAGC-3′ (Guérillon *et al*, 2020); PIAS2: 5′-CUUGAAUAUUACAUCUUUAUT-3′ (Galanty *et al*, 2009); PIAS3: 5′-CCCUGAUGUCACCAUGAAATT-3′ (Galanty *et al*, 2009); ZATT: 5′-CAGAAUUCAGGACACAAAUU-3′ (Borgermann *et al*, 2019); DNMT1: 5′-CGGUGCUCAUGCUUACAAC-3′; PIAS4 #1: 5′-GGAAUAAGAGUGGACUGAATT-3′ and PIAS4 #3: 5′- AGCUGCCGUUCUUUAAUAUTT-3′ (Fig EV2L); PIAS4 (3′UTR): 5′-UAGCCACAGAUGUGUUGUAUU-3′ (Fig EV2N); RNF4 #1: 5′-GAA UGGACGUCUCAUCGUUTT-3′ and RNF4 #2: 5′-GAAUGGACGUCU CAUCGUU-3′ (Fig EV1C); RNF111 #1: 5′-AGAAGGAAAUGAAUG

GUAATT-3′ and RNF111 #2: HSS182646 (Thermo Scientific) (Fig EV1D; a mix of siRNA #1 and #2 was used in all experiments).

## Immunochemical methods, chromatin fractionation and antibodies

Immunoblotting was performed as previously described (Poulsen *et al*, 2012). To prepare cell extracts, cells were lysed in EBC buffer supplemented with protease and phosphatase inhibitors on ice for 20 min, and lysates were cleared by centrifugation (16,300 × *g*, 20 min). For immunoprecipitation, cells were lysed in denaturing buffer (20 mM Tris, pH 7.5; 50 mM NaCl; 1 mM EDTA; 0.5% NP-40; 0.5% SDS, 0.5% sodium deoxycholate; 1 mM DTT) supplemented with protease and phosphatase inhibitors, followed by sonication. Lysates were then cleared by centrifugation (16,300 × *g*, 20 min). Cleared lysates were incubated with GFP-trap agarose (Chromotek) overnight with constant agitation at 4°C. After extensive washing of the beads, GFP-tagged proteins were eluted by boiling in 2× Laemmli sample buffer for 5 min. For chromatin fractionation, cells were first lysed in buffer 1a (10 mM Tris, pH 8.0; 10 mM KCl; 1.5 mM MgCl$_2$; 0.34 M sucrose; 10% glycerol; 0.1% Triton X-100) supplemented with protease and phosphatase inhibitors on ice for 5 min, followed by centrifugation (2,000 × *g*, 5 min) to recover the soluble proteins. Pellets were then washed extensively twice in buffer 1b (buffer 1a supplemented with 500 mM NaCl), followed by resuspension in buffer 2 (50 mM Tris, pH 7.5; 150 mM NaCl; 1% NP-40; 0.1% SDS; 1 mM MgCl$_2$; 125 U/ml benzonase) supplemented with protease and phosphatase inhibitors. Lysates were incubated in thermomixer (37°C, 1,000 rpm, 15 min), and solubilized chromatin-bound proteins were obtained by centrifugation (16,300 × *g*, 10 min).

Antibodies to human proteins used in this study included: actin (clone C4, MAB1501, Merck (1:20,000 dilution), RRID:AB_2223041); ATM (ab78, Abcam (1:1,000), RRID:AB_306089); ATM-pS1981 (4526, Cell Signaling Technology (1:1,000), RRID:AB_2062663); CDC27 (sc-9972, Santa Cruz, RRID:AB_627228); CDK1-pY15 (ab47594, Abcam (1:1,000), RRID:AB_869073); CHK1 (sc-8408, Santa Cruz (1:500), RRID:AB_627257); CHK1-pS317 (2344S, (Cell Signaling Technology (1:1,000), RRID:AB_331488); cyclin A (sc-751, Santa Cruz (1:1,000), RRID:AB_631329); cyclin B (610220, BD Bioscience (1:1,000), RRID:AB_397617); GFP (11814460001 (clones 7.1 and 13.1), Roche (1:1,000), RRID:AB_390913; ab6556, Abcam (1:5,000), RRID:AB_305564); histone H2AX (2595S, Cell Signaling Technology (1:1,000), RRID:AB_10694556); histone γ-H2AX (05-636, Millipore (1:1,000), RRID:AB_309864); histone H3 (ab1791, Abcam (1:50,000), RRID:AB_302613); histone H3-pS10 (06-570, Millipore (1:1,000), RRID:AB_310177); KAP1 (A300-274A, Bethyl Laboratories (1:1,000), RRID:AB_185559); KAP1-pS824 (A300-767A, Bethyl (1:1,000), RRID:AB_669740); NF-κB p65 (sc-372, Santa Cruz (1:2,000); RRID:AB_632037); RNF4 (Maure *et al*, 2016) (1:6000)); PIAS4 (sc-166706, Santa Cruz (1:1,000); RRID: AB_2164374); SUMO2/3 (ab3742, Abcam (1:1,000), RRID:AB_304041; ab81371 (8A2), Abcam (1:200), RRID:AB_1658424); ubiquitin (sc-8017 (P4D1), Santa Cruz (1:1,000), RRID:AB_628423); ubiquitin (K48) (05-1307, Millipore (1:1,000), RRID:AB_1587578). Polyclonal sheep antibody to DNMT1 was raised against full-length recombinant human DNMT1 purified from bacteria (Borgermann *et al*, 2019).

Antibodies to *Xenopus* proteins used in this study and previously described include the following: M.HpaII (Larsen *et al*, 2019); ORC2

(Fang & Newport, 1993); PARP1 (Ryu *et al*, 2010); PIAS4 (Azuma *et al*, 2005); PSMA1 (Larsen *et al*, 2019); PSMA3 (Larsen *et al*, 2019). RNF4 antibody was raised against the following peptides: RNF4 CT-L Ac-CRKKLNHKQYHPIYI-OH and RNF4 CT-S Ac-CRKKLNNKQYHPIYV-OH (New England Peptide). The following commercially available antibodies were used: Histone H3 (9715S, Cell Signaling Technology), PAR (83732, Cell Signaling Technology).

## Immunofluorescence and high-content image analysis

Cells were pre-extracted on ice in stringent pre-extraction buffer (10 mM Tris–HCl, pH 7.4; 2.5 mM MgCl$_2$; 0.5% NP-40; 1 mM PMSF) for 8 min and then in ice-cold PBS for 2 min prior to fixation with 4% formaldehyde for 15 min. If not pre-extracted, cells were subjected to permeabilization with PBS containing 0.2% Triton X-100 for 5 min prior to blocking. Coverslips were blocked in 10% BSA and incubated with primary antibodies for 1 h at room temperature, followed by staining with secondary antibodies and DAPI (Alexa Fluor; Life Technologies) for 1 h at room temperature. Coverslips were mounted in MOWIOL 4-88 (Sigma). Manual image acquisition was performed with a Leica AF6000 wide-field microscope (Leica Microsystems) equipped with HC Plan-Apochromat 63×/1.4 oil immersion objective, using standard settings and LAS X software (Leica Microsystems). Raw images were exported as TIFF files, and if adjustments in image contrast and brightness were applied, identical settings were used on all images of a given experiment. For automated image acquisition, images were acquired with Olympus IX-81 wide-field microscope equipped with an MT20 Illumination system, Olympus UPLSAPO 20×/0.75 NA objective, and a digital monochrome Hamamatsu C9100 CCD (charge-coupled device) camera was used. Automated and unbiased image analysis was carried out with the ScanR analysis software. Data were exported and processed using Spotfire (TIBCO Software Inc.).

## Quantification of mRNA levels by RT–qPCR

RNA was purified from cells (RNeasy kit, Qiagen), and cDNA was generated by PCR with reverse transcription (iScript cDNA Synthesis Kit, Bio-Rad) according to the manufacturer's instructions. Real-time quantitative PCR was performed using the Stratagene Mx3005P System and Brilliant III Ultra-Fast SYBR Green QPCR Master Mix (Agilent). *GAPDH* mRNA levels were used as a control for normalization. For amplification of the indicated cDNAs, the following primers were used: GAPDH (forward): 5′-CAGAACATCATCCC TGCCTCTAC-3′; GADPH (reverse): 5′-TTGAAGTCAGAGGAGACC ACCTG-3′; RNF111 (forward): 5′-TGCATCCTCACTTGGCCCAT-3′; RNF111 (reverse): 5′-TCAGTTCCTCAAAATTGCCCCTG-3′.

## Clonogenic survival assays

Approx. 400 cells were plated per 60-mm plate and treated with 5-azadC for 24 h or formaldehyde for 30 min. Cells were then washed with PBS twice, and fresh medium was replenished. Colonies were fixed and stained after approximately 2 weeks with cell staining solution (0.5% w/v crystal violet, 25% v/v methanol). The number of colonies were quantified using a GelCount™ (Oxford Optronix) colony counter.

## Flow cytometry

Cells were fixed in ice-cold 80% ethanol, and DNA was stained with propidium iodide (0.1 mg/ml) containing RNase (20 μg/ml) for 30 min at 37°C. Flow cytometry analysis was performed on a FACS Calibur (BD Biosciences) using CellQuest Pro software (version 6.0; Becton Dickinson). The data were analysed using FlowJo software (version 10.6).

## Live cell imaging

Cells were seeded onto an Ibidi dish (Ibidi) the day before acquisition. Medium were changed to Leibovitz's L15 medium (Life Technologies) supplemented with 10% FBS and SiR-DNA (100 nM, Spirochrome) prior to filming. Live cell imaging was performed on a Deltavision Elite system using a 40× oil objective (GE Healthcare). Images (DIC and Cy5 channels) were acquired every 5 min for 18 h. Three z-stacks of 5 μm were imaged. SoftWork software (GEHealthcare) was used for data analysis.

## *In vitro* STUbL assays

HeLa/GFP-DNMT1 cells treated or not with 5-azadC were lysed in denaturing buffer supplemented with protease and phosphatase inhibitors, and lysates were sonicated and cleared by centrifugation. GFP-tagged DNMT1 was then purified on GFP-Trap agarose (Chromotek) followed by extensive washing in denaturing buffer. The beads were equilibrated in ubiquitylation assay buffer (50 mM Tris pH 7.5; 150 mM NaCl; 5 mM $MgCl_2$; 2 mM NaF; 2 mM ATP; 6 mM DTT; 0.1% NP-40) and incubated with recombinant proteins (100 nM E1, 500 nM E2 (UbcH5a), 20 μM HA-ubiquitin, and 0.55 μM RNF4 with shaking at 37°C for 1 h. Bound material was washed in denaturing buffer, eluted by boiling in 2× Laemmli sample buffer for 5 min and analysed by immunoblotting.

## Xenopus egg extracts

Egg extracts were prepared using *Xenopus laevis* (Nasco Cat #LM0053MX, LM00715MX). All experiments involving animals were approved by the Danish Animal Experiments Inspectorate and conform to relevant regulatory standards and European guidelines.

Preparation of *Xenopus* egg extracts was performed as described previously (Lebofsky *et al*, 2009; Sparks & Walter, 2019). For high-speed supernatant (HSS) preparation, 6 female frogs (Nasco) were primed by injection with 80 IU of human chorionic gonadotropin (hCG, Sigma). 2–7 days after priming, frogs were injected with 625 IU of hCG and placed in individual tanks containing 100 mM NaCl. 18-20 h post injection, eggs were collected and used for extract preparation. Eggs were first dejellied in cysteine buffer for 7 min (2.2% cysteine-HCl, pH 7.7), washed three times in 0.5× MMR buffer (final concentration: 50 mM NaCl, 1 mM KCl, 0.25 mM $MgSO_4$, 1.25 mM $CaCl_2$, 2.5 mM HEPES, 0.05 mM EDTA, pH 7.8) and washed three times in ELB sucrose buffer (2.5 mM $MgCl_2$, 50 mM KCl, 10 mM HEPES, 250 mM sucrose, 1 mM DTT, 50 μg/ml cycloheximide, pH 7.8). Eggs were packed for 1 min at 176 × *g* and crushed for 20 min at 20,000 × *g* in a swing bucket rotor at 4°C in the presence of cytochalasin B (2.5 μg/ml), aprotinin (5 μg/ml) and leupeptin (5 μg/ml). Crude interphase extract was

recovered post-centrifugation and spun in ultracentrifuge for 90 min at 260,000 × *g* at 2°C following addition of cycloheximide (50 μg/ml), DTT (1 mM), aprotinin (10 μg/ml), leupeptin (10 μg/ml) and cytochalasin B (5 μg/ml). Following centrifugation, the lipid layer on top was removed. The soluble HSS was harvested, snap frozen in 33 μl aliquots and stored at −80°C. For nucleoplasmic egg extract (NPE), 20 female frogs were injected and the crude interphase extract was prepared in the same manner as for HSS. Once collected, the crude interphase extract was supplemented with cycloheximide (50 μg/ml), DTT (1 mM), aprotinin (10 μg/ml), leupeptin (10 μg/ml), cytochalasin B (5 μg/ml) and nocodazole (3.3 μg/ml). The extract was spun at 20,000 × *g* at 4°C for 10 min. The lipid layer on top was removed and the interphase extract decanted to a new tube. The interphase extract was supplemented with ATP (final concentration: 2 mM), phosphocreatine (20 mM) and creatine phosphokinase (5 μg/ml), and nuclear assembly reactions were initiated by adding demembranated sperm chromatin to a final concentration of 4,400/μl. The nuclear assembly reaction was incubated at room temperature for 60–85 min and then spun for 2 min at 20,000 × *g* in a swing bucket rotor. The nuclear layer on top was recovered and then spun in a swinging bucket rotor at 260,000 × *g* at 2°C for 30 min. Lipids on top were removed and the clear soluble NPE was harvested. NPE (10 μl) aliquots were snap-frozen and kept at −80°C.

CSF-arrested whole egg extracts were prepared as previously described (Hannak & Heald, 2006) but with the following modification: The low speed supernatant extract recovered from the Hannak *et al* protocol was transferred to a new tube and spun at 260,000 × *g* at 2°C for 2 h. The soluble fraction was collected and snap frozen in single use 20 μl aliquots.

For replication-independent reactions, plasmid DNA was incubated at 6 ng/μl in the indicated extract. NPE was supplemented with 4 mM DTT, 20 mM phosphocreatine, 2 mM ATP, 5 μg/ml creatine phosphokinase; HSS and CSF-arrested extract were supplemented with nocodazole (3 μg/ml), 20 mM phosphocreatine, 2 mM ATP, 5 μg/ml creatine phosphokinase. Plasmid DNA was recovered via DPC pull-down as described below.

As previously described (Duxin *et al*, 2014) for DNA replication, plasmids were first incubated in HSS (7.5 ng DNA/μl HSS) for 20-30 min at room temperature to license the DNA. Two volumes of NPE were then added to 1 volume of licensing reaction to initiate replication. To block *de novo* SUMOylation, ML-792 SUMO-E1 inhibitor was added to extracts to a final concentration of 50 μM. To block *de novo* ubiquitylation MLN-7243 was added to a final concentration of 133 μM. For DNA labelling, reactions were supplemented with [α-32P]dATP. To analyse plasmid replication intermediates, 1 μl of each reaction was added to 5 μl of replication stop solution A (5% SDS, 80 mM Tris pH 8.0, 0.13% phosphoric acid, 10% Ficoll) supplemented with 1 μl of Proteinase K (20 mg/ml) (Roche). Samples were incubated for 1 h at 37°C prior to separation by 0.9% native agarose gel electrophoresis and visualization using a phosphorimager (Lebofsky *et al*, 2009).

## Immunodepletion and add-back experiments

To immunodeplete PIAS4, RNF4 or PSMA1 from *Xenopus* egg extracts, one volume of Protein A Sepharose Fast Flow (PAS) (GE Health Care) was mixed with either five volumes of PIAS4 serum

antibody, five volumes of affinity purified RNF4 peptide antibody (1 mg/ml) or 10 volumes of affinity purified PSMA1 peptide antibody (1 mg/ml) and incubated overnight at 4°C. The beads were subsequently washed twice with 500 µl PBS, once with ELB-sucrose (10 mM HEPES pH 7.7, 50 mM KCl, 2.5 mM $MgCl_2$, and 250 mM sucrose), twice with ELB supplemented with 0.5 M NaCl and twice with ELB-sucrose. Five volumes of NPE or CSF arrested extract was then depleted by mixing with one volume of antibody-bound beads and incubated at room temperature for 15 min. This was repeated once for PIAS4, twice for RNF4 and three times for PSMA1.

For rescue experiments, PIAS4 WT or SIM1/2* were added to depleted extracts to a final concentration of 10 ng/µl in the reaction. RNF4 WT or a catalytically inactive C130A mutant were added to a final concentration of 7 ng/µl or five times this amount where indicated (5xRNF4).

## Preparation of DNA constructs

To generate p4xDPC and p4xDPC$^{SUMO}$, pJLS3 (Sparks et al, 2019) was nicked with Nt.BbvCI (New England Biolabs) and ligated with the following oligo containing a fluorinated cytosine: 5′-TCAGCATC[C5-Fluoro-dC]GGTAGCTACTCAATC[C5-Fluoro-dC]GGTACC-3′ and subsequently crosslinked to M.HpaII-His$_6$ or SUMOΔGG-M.HpaII-His$_6$, respectively, as previously described (Duxin et al, 2014). Briefly, the fluorinated plasmid DNA was gel purified and mixed with M.HpaII-His$_6$ or SUMOΔGG-M.HpaII-His$_6$ in reaction buffer (50 mM Tris–HCl pH 7.5, 5 mM 2-mercaptoethanol, 10 mM EDTA) supplemented with 100 µM of S-adenosylmethionine (NEB) for 12 hr at 37°C.

pDPC$^{ssDNA}$ and pDPC were previously described in (Larsen et al, 2019). p2xDPC$^{Leads}$ was previously described in (Larsen et al, 2019) as pDPC$^{2xLeads}$. p2xDPC$^{SUMOLeads}$ and pDPC$^{SUMOssDNA}$ were generated similarly to p2xDPC$^{Leads}$ and pDPC$^{ssDNA}$, but plasmid DNA was crosslinked to SUMOΔGG-M.HpaII-His$_6$ instead of M.HpaII-His$_6$.

## Protein expression and purification

M.HpaII-His6 was expressed and purified as previously described (Duxin et al, 2014). Briefly, pHpaII-Avitag-His$_6$ was transformed in T7 Express Competent E. coli cells (NEB), which were cultured in the presence of 100 µg/ml ampicillin until the OD$_{600}$ reached 0.7. The culture was supplemented with 0.5 mM IPTG for 3 h, collected by centrifugation and resuspended in 15 ml Lysis Buffer (20 mM Tris pH 8.5, 500 mM KCl, 10% glycerol, 10 mM imidazole and protease inhibitors (Roche)). Cells were lysed by sonication and cleared by centrifugation at $20,000 \times g$ for 30 min. Cleared lysate was applied onto Ni-NTA resin (Qiagen). The resin was washed with 25 ml of Lysis Buffer containing 30 mM imidazole and the protein eluted with Elution Buffer (20 mM Tris pH 8.5, 100 mM KCl, 10% glycerol and 250 mM imidazole). Eluate was dialysed overnight in Storage Buffer (20 mM Tris pH 8.5, 100 mM KCl, 1 mM DTT, 30% glycerol) and protein aliquots snap frozen and kept at −80°C. SUMOΔGG-M.HpaII was expressed and purified in the same way.

LacI-biotin protein was purified from T7 Express Competent cells (NEB) (Duxin et al, 2014). Briefly, pET11a-LacI and pBirAcm (Avidity) were co-transformed and cells cultured in the presence of

100 µg/ml ampicillin and 34 µg/ml chloramphenicol at 37°C until OD$_{600}$ reached 0.6. The culture was supplemented with 1 mM IPTG and 50 µM biotin for 2 h. Cells were collected by centrifugation and resuspended in Buffer 1 (50 mM Tris pH 7.5, 5 mM EDTA, 100 mM NaCl, 10% sucrose, 1 mM DTT, protease inhibitors (Roche), 0.2 mg/ml lysozyme (Sigma), 0.1% Brij 58) and rotated for 30 min at room temperature. The cell lysate was pelleted by centrifugation for 60 min at $20,000 \times g$, and the insoluble pellet was resuspended in 10 ml of Extraction Buffer (50 mM Tris pH 7.5, 5 mM EDTA, 1 M NaCl, 30 mM IPTG, 1 mM DTT and protease inhibitors). The resuspended pellet was homogenized by sonication and pelleted again for 60 min at $20,000 \times g$. The supernatant was collected, and 1% polymin P was added to 0.045%. Lysate was rotated for 30 min at 4°C and pelleted at $20,000 \times g$ for 20 min. The supernatant was transferred to a new tube, and ammonium sulphate was added to a final saturation of 37% followed by rotation for 30 min at 4°C. The pellet was recovered and resuspended in 2 ml of Wash Buffer (50 mM Tris pH 7.5, 1 mM EDTA, 100 mM NaCl, 1 mM DTT and protease inhibitors). The resuspension was applied to a column containing 1 ml of softlink avidin resin and incubated for 1 h at 4°C. The column was washed with 15 ml of Wash Buffer, and the protein eluted with Elution buffer (50 mM Tris pH 7.5, 1 mM EDTA, 100 mM NaCl, 1 mM DTT and 5 mM biotin). Protein was dialyzed overnight with Dialysis Buffer (50 mM Tris pH 7.5, 1 mM EDTA, 150 mM NaCl, 1 mM DTT and 30% glycerol) and stored at −80°C.

Xenopus RNF4.L cDNA (Horizon) was cloned into pET28b using primers 5′-ATGCCATATGACAGCAGTGACTG-3′ and 5′-ACTGAAG CTTTCATATATATATAGGGTGATATTG-3′ and mutations were introduced via QuikChange Site-directed mutagenesis. RNF4 WT or C130A were transformed into T7 Express Competent E.coli cells (NEB) and grown in the presence of 50 µg/ml of kanamycin. Expression was induced with 100 µM IPTG for 4 h at 37°C. One liter of culture was harvested and lysed in 25 ml lysis buffer, pH 7.4 (50 mM Tris–HCl, 300 mM NaCl, 2 mM $MgCl_2$, 1 mM DTT, 1 tablet complete/50 ml) via sonication. The lysate was supplemented with 25 U/ml of benzonase and incubated for 20 min at room temperature and centrifuged subsequently at $22,000 \times g$ for 1 h at 4°C. The supernatant was bound to 2 ml bead slurry of Ni-NTA resin (Qiagen) for 2 h rotating at 4°C. The beads were washed with 60 ml wash buffer (50 mM Tris–HCl, 300 mM NaCl, 2 mM $MgCl_2$, 1 mM DTT, 0.1% Triton X-100) in total. The protein was eluted from the resin with elution buffer (50 mM Tris–HCl, 300 mM NaCl, 2 mM $MgCl_2$, 1 mM DTT, 10% glycerol, 300 mM imidazole) and dialyzed with 50 mM Tris–HCl, 300 mM NaCl, 2 mM $MgCl_2$, 1 mM DTT, 10% glycerol overnight and with fresh dialysis buffer for two additional h the following day and stored at −80°C.

Expression and purification of human RNF4 proteins (WT, *RING and *SIM mutants) were done as previously described (Tatham et al, 2008; Plechanovová et al, 2011).

Xenopus PIAS4 (PIASγ) WT and SIM1/2* were purified as previously described (Azuma et al, 2005). Briefly, the protein was expressed in E. coli and purified through a Talone-sepharose FF column followed by a cation exchange column (SP-sepharose). The peak elution was further separated by anion exchange (Q-sepharose). The final protein preparation was dialyzed into 150 mM NaCl, 25 mM HEPES, 5% glycerol and 1 mM DTT (pH 7.8).

## DPC pull-downs

DPC pull-downs were performed as previously described (Larsen *et al*, 2019). Briefly, streptavidin-coupled magnetic beads (Dynabeads M-280, Invitrogen; 5 µl per pull-down) were washed twice with 50 mM Tris pH 7.5, 150 mM NaCl, 1mM EDTA pH 8.0, 0.02% Tween-20. Biotinylated LacI was added to the beads (1 pmol per 5 µl of beads) and incubated at room temperature for 40 min. The beads were then washed four times with DPC pull-down buffer (20 mM Tris pH 7.5, 150 mM NaCl, 2 mM EDTA pH 8, 0.5% IPEGAL-CA630) and then stored in the same buffer on ice until needed. At the indicated times, 5 µl of reaction was withdrawn and stopped in 300 µl of DPC pull-down buffer on ice. After all time points were taken, 5 µl of LacI-coated streptavidin Dynabeads was added to each sample and allowed to bind for 30–60 min at 4°C with rotation. The beads were subsequently washed four times with DPC pull-down buffer and then twice with Benzonase buffer (20 mM Tris pH 7.5, 150 mM NaCl, 2 mM $MgCl_2$, 0.02% Tween-20) before being resuspended in 15 µl Benzonase buffer containing 1 µl Benzonase (Merck Millipore). Samples were incubated for 1 h at 37°C to allow for DNA digestion and DPC elution, after which the beads were pelleted and the eluate was mixed with 2× Laemmli sample buffer for subsequent western blotting analysis. For treatment with Ulp1 and Usp2, 5 µl of reaction was withdrawn at the indicated time points and stopped in 300 µl of DPC pull-down buffer on ice. After all of the time points were taken, 5 µl of LacI-coated streptavidin Dynabeads was added to each sample and allowed to bind for 30–60 min at 4°C rotating. The beads were subsequently washed four times with DPC pull-down buffer and then once with Benzonase buffer (20 mM Tris pH 7.5, 150 mM NaCl, 2 mM $MgCl_2$, 0.02% Tween-20) and once with DUB reaction buffer (500 mM Tris 7.5, 500 mM NaCl, 50 mM DTT) supplemented with 0.02% Tween-20 before being resuspended in 15 µl DUB reaction buffer containing 1 µl Ulp1 and/or 1 µl Usp2. Samples were incubated for 45 min at 37°C and subsequently washed once with Benzonase buffer (20 mM Tris pH 7.5, 150 mM NaCl, 2 mM $MgCl_2$, 0.02% Tween-20) before being resuspended in 15 µl Benzonase buffer containing 1 µl Benzonase and incubated for 1 h at 37°C, after which the beads were pelleted and the eluate was mixed with 2× Laemmli sample buffer for subsequent immunoblotting analysis.

## Chromatin spin-down

Demembranated *Xenopus* sperm chromatin was prepared as described (Sparks & Walter, 2019) and stored at −80°C at a concentration of 100,000 sperm chromatin/µl (320 ng/µl). Where indicated, sperm chromatin was diluted to 50,000 sperm chromatin/µl in ELB buffer (10 mM HEPES pH 7.7, 50 mM KCl, 2.5 mM $MgCl_2$, and 250 mM sucrose) and irradiated with 2000 J/$m^2$ of UV-C. Sperm chromatin was added at a final concentration of 16 ng/µl to one volume of HSS and two volumes of NPE that were premixed. At the indicated time points, 8 µl of reaction was stopped in 60 µl of ELB buffer supplemented with 0.2% Triton X. The mixture was carefully layered on top of a sucrose cushion (10 mM HEPES pH 7.7, 50 mM KCl, 2.5 mM $MgCl_2$ and 500 mM sucrose) and spun for 1 min at 6,800 × *g* in a swing bucket centrifuge at 4°C. The chromatin pellet was carefully washed twice with

200 µl of ice-cold ELB buffer and resuspended in 2× Laemmli buffer.

## *In vitro* DPC SUMOylation

DPCs were SUMOylated by incubation overnight at 4°C with 100 ng Sae1/Uba2, 300 ng Ubc9 and 3 µg SUMO2 per 10 µl reaction in 1x Buffer S (50 mM HEPES pH 7.5, 100 mM NaCl, 10 mM $MgCl_2$, 100 nM DTT, 2 mM ATP). Where indicated, PIAS4 was added to the reaction at 80 ng/µl.

## Extract exchange

Plasmid DNA was incubated in NPE for 60 min. SUMOylated DPCs were then recovered via DPC pull-down, beads were washed with ELB sucrose buffer (2.5 mM $MgCl_2$, 50 mM KCl, 10 mM HEPES, 250 mM sucrose) supplemented with 0.02% Tween-20, dried and whole egg CSF-arrested Xenopus egg extract was added to a final DNA concentration of 6 ng/µl and incubated rotating at room temperature. At the indicated time points, 5 µl was withdrawn and stopped in DPC pull-down buffer (20 mM Tris pH 7.5, 150 mM NaCl, 2 mM EDTA pH 8, 0.5% IPEGAL-CA630). After the last time point, the samples were incubated for 30 min rotating at 4°C and from here on processed as described above for the DPC pull-down procedure.

## Imaging of sperm chromatin

Sperm chromatin was incubated at 16 ng/µl in CSF-arrested extracts. CDK inhibitor R547 was added where indicated at a final concentration of 10 µM. Samples were taken at 60 min, fixed in Hoechst stain solution (8 µg/ml Hoechst, 7.4% formaldehyde, 200 mM sucrose, 10 mM HEPES pH 7.6). Images were obtained with a Leica AF6000 wide-field microscope (Leica Microsystems) equipped with HC Plan-Apochromat 63×/1.4 oil immersion objective, using standard settings and LAS X software (Leica Microsystems).

## CHROMASS

CHROMASS experiments were performed as previously described (Räschle *et al*, 2015). Briefly, isolated sperm chromatin was treated with 2,000 J/$m^2$ of UV-C. Each reaction was performed in quadruplicate. The sperm chromatin was then incubated at a final concentration of 16 ng/µl in non-licensing extracts in the presence or absence of Talazoparib (10 µM) and SUMOi (50 µM). Reactions were stopped after 30 min. Specifically, 10 µl of reaction was stopped with 60 µl of ELB buffer supplemented with 0.2% Triton X, and chromatin spin down performed as described above. The chromatin pellet was then resuspended in 50 µl denaturation buffer (8 M urea, 100 mM Tris–HCl, pH 8.0) and transferred to a new low binding tube. Cysteines were reduced (1 mM DTT for 15 min at RT) and alkylated (0.55 M chloroacetamide for 40 min at RT protected from light). Proteins were first digested with 0.5 µg Lys-C (2.5 h at RT) and then with 0.5 µg trypsin at 30°C overnight. Peptides were acidified with 10% trifluoroacetic acid (pH < 4), followed by addition of 400 mM NaCl and purified by stage tipping (C18 material). For this, stage tips were first activated in 100% methanol, then equilibrated in 80% acetonitrile/10% formic acid, and finally washed twice in 0.1% formic acid. Samples were loaded on the equilibrated

stage tips and washed twice with 50 µl 0.1% formic acid. StageTip elution was performed with 80 µl of 25% acetonitrile in 0.1% formic acid, eluted samples were dried to completion in a SpeedVac at 60°C, dissolved in 10 µl 0.1% formic acid and stored at −20°C until MS analysis.

## MS acquisition

All CHROMASS samples were analysed on an EASY-nLC 1200 system (Thermo) coupled to a Q Exactive™ HF-X Hybrid Quadrupole-Orbitrap™ mass spectrometer (Thermo). Separation of peptides was performed using 15-cm columns (75 µm internal diameter) packed in-house with ReproSil-Pur 120 C18-AQ 1.9 µm beads (Dr. Maisch). The columns were heated to 40°C using a column oven (Sonation). Elution of peptides from the column was achieved using a gradient ranging from buffer A (0.1% formic acid) to buffer B (80% acetonitrile in 0.1% formic acid), at a constant flow rate of 250 nl/min. Gradient length was 100 min per sample, including ramp-up and wash-out, with an analytical gradient of 76 min ranging from 8 to 28% buffer B. Ionization of peptides was performed using a NanoSpray Flex™ ion source (Thermo), with spray voltage set to 2 kV, ion transfer tube temperature to 275°C and RF funnel level to 40%. All samples were measured as two technical replicates using 5 µl of the sample per injection, with differences outlined below. MS RAW data files with a "b" appended to the file name correspond to those analysed using alternative settings. Measurements were performed with a full scan range of 300–1,750 $m/z$, MS1 resolution of 60,000 (or alternatively 120,000), MS1 AGC target of 3,000,000 and MS1 maximum injection time of 60 ms. Precursors with charges 2–6 were selected for fragmentation using an isolation width of 1.3 $m/z$ and fragmented using higher-energy collision disassociation (HCD) with a normalized collision energy of 25. Precursors were excluded from re-sequencing by setting a dynamic exclusion of 100 s. MS2 AGC target was set to 200,000, minimum MS2 AGC target to 20,000, MS2 maximum injection time to 55 ms, MS2 resolution to 30,000 and loop count to 14 (or alternatively 12).

## MS data analysis

All MS RAW data were analysed using the freely available MaxQuant software (Cox & Mann, 2008), version 1.5.3.30. Default MaxQuant settings were used, with exceptions specified below. For generation of theoretical spectral libraries, the *Xenopus laevis* FASTA database was downloaded from UniProt on the 10th of February, 2018. In silico digestion of proteins to generate theoretical peptides was performed with trypsin, allowing up to 3 missed cleavages. Maximum variable modifications per peptide were reduced to 3. Label-free quantification (LFQ) was enabled (Cox *et al*, 2014), with "Fast LFQ" disabled, and "LFQ min. ratio count" set to 3. Second peptide search was enabled. Matching between runs was enabled, with an alignment window of 20 min and a match time window of 1 min. Stringent MaxQuant 1% FDR was applied at the PSM and protein levels (default).

## MS data annotation and quantification

The *Xenopus laevis* FASTA database downloaded from UniProt lacked comprehensive gene name annotation. Missing or uninformative gene names were, when possible, semi-automatically curated by drawing informative gene names from UniProt, Xenbase, the Session *et al* database (Session *et al*, 2016) or RefSeq (via Xenbase), otherwise InterPro annotations were used. Quantification of the MaxQuant output files was performed using the freely available Perseus software (Tyanova *et al*, 2016), v1.5.5.3. For quantification, all protein LFQ intensity values were $\log_2$ transformed, and proteins were filtered for valid (i.e. non-zero) values in 4 out of 4 replicates ($n = 4/4$) in at least one experimental condition. Missing values were imputed below the global experimental detection limit at a downshift of 1.8 and a randomized width of 0.3 (in $\log_2$ space; Perseus default). The statistical significance of differences was evaluated using two-tailed Student's *t*-testing, with permutation-based FDR-control applied at an s0 value ("fudge factor") of 0.5. A first round of quantification was performed where all UV-treated conditions were individually compared to the control, and only proteins with a positive ratio and an FDR-adjusted *q*-value of < 10% in at least one of these comparisons were subjected to two-sample testing within UV-treated conditions. Both *P*-values and FDR-adjusted *q*-values are reported in Dataset EV1.

# Data availability

The mass spectrometry proteomics data (Dataset EV1) have been deposited to the ProteomeXchange Consortium via the PRIDE partner repository with the dataset identifier PXD021947 (https://www.ebi.ac.uk/pride/archive/projects/PXD021947). All other data supporting the findings of this study are available within the article and supplementary information.

Expanded View for this article is available online.

## Acknowledgements

We thank members of the Duxin and Mailand laboratories for helpful discussions. We also thank the Novo Nordisk Foundation Center for Protein Research protein production facility for purifying PIAS4 and LacI-biotin. This work was supported by grants from Novo Nordisk Foundation (grants no. NNF14CC0001 and NNF18OC0030752), European Research Council (ERC, grant agreement no. 616236 (DDRegulation) and no. 715975 (DPC_REPAIR)), and Lundbeck Foundation (grant no. R223-2016-281). J.C.Y.L. and U.K. are supported by Novo Nordisk Foundation (grants no. NNF18CC0033876 and NNF17CC0026748). J.C.Y.L. is supported by the Croucher Foundation. L.A. is a Marie Skłodowska-Curie Fellow (IF-Grant Number: 798560-DNAProteinCrossRep). The R.T.H. lab is supported by Wellcome (217196/Z/19/Z) and Cancer Research UK (C434/A21747). C.G. was supported by the Danish Cancer Society (grant no. R146-RP11394).

## Author contributions

Conceptualization: JCYL, UK, JPD and NM; Methodology: JCYL, UK, NB, DHG, IAH, JPD and NM; Investigation: JCYL, UK, NBL, DHG, IAH, LA, IG, CG; Resources: PH, EB, RTH and YA; Writing – Original Draft: JPD and NM; Writing – Review and Editing: All authors; Supervision, MLN, JPD and NM; Project Administration: JPD and NM; Funding Acquisition: LA, CG, RTH, JPD and NM.

## Conflict of interest

The authors declare that they have no conflict of interest.

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
