## [Review Process File · The EMBO Journal]

Mechanism and function of DNA replication-independent DNA-protein crosslink repair via the SUMO-RNF4 pathway

Niels Mailand, Julio Liu, Ulrike Kühbacher, Nicolai Larsen, Nikoline Borgermann, Dimitriya Garvanska, Ivo Hendriks, Leena Ackermann, Peter Haahr, Irene Gallina, Claire Guérillon, Emma Branigan, Ronald Hay, Yoshiaki Azuma, Michael Nielsen, and Julien Duxin

DOI: [10.15252/embj.2020107413](https://doi.org/10.15252/embj.2020107413)

Corresponding author(s): Niels Mailand (niels.mailand@cpr.ku.dk), Julien Duxin (julien.duxin@cpr.ku.dk)

Review Timeline:

Submission Date:	29th Nov 20
Editorial Decision:	18th Dec 20
Revision Received:	19th May 21
Editorial Decision:	11th Jun 21
Revision Received:	3rd Jul 21
Accepted:	12th Jul 21

Editor: Hartmut Vodermaier

Transaction Report:

Thank you for submitting your manuscript on STUbL pathway involvement in DPC removal. It has now been reviewed by three expert referees, whose comments are copied below. As you will see, at least two referees appreciate the principle interest and importance of these findings, but all three reviewers also raise a number of significant concerns that would clearly need to be addressed before eventual publication. In light of the overall interest and support from two of the reviewers, I would nevertheless like to give you an opportunity to respond to the criticism by way of a revised manuscript. For such a revision, it will be important to not only moderate some claims/conclusions, but also to strengthen various results, especially the genetic/epistatic evidence for the involvement of particular SUMO and ubiquitin E3s, and those extending to non-Dnmt1 DPCs.

REFEREE REPORTS

Referee #1:

GENERAL SUMMARY AND SIGNIFICANCE

The manuscript by Liu et al reports a follow-up study to their recent publication showing that covalent DNA-protein crosslinks undergo SUMOylation-dependent but replication-independent

degradation by the proteasome and the metalloprotease ACRC (Borgermann et al EMBO 2019). The authors use cell-based assays and experiments conducted in *X. laevis* egg extracts to identify the PIAS4 SUMO E3 ligase as being responsible for DPC SUMOylation and the STUBL RNF4 for subsequent ubiquitylation. Similar to what has just been suggested for topoisomerase-DPCs by the Pommier lab (Sun et al., 2020). The authors find that DPC SUMOylation is unrelated to replication-coupled repair of DPCs, but is crucial for the removal of protein adducts prior to mitosis in an RNF4-dependent but replication-independent manner.

The manuscript provides new insights regarding the identity of the enzymes modifying DPCs for repair, but the general concept of SUMO-dependent degradation of DPCs has been proposed previously (Borgermann et al, EMBO 2019, Fielden et al, Nat Comm 2020). The authors analyse the fate of trapped DNMT1 in cells but draw general conclusion regarding the repair of DPCs. Thus, claims in title/abstract should either be reduced to DNMT1-DPCs or the cellular data has to be extended to aldehyde-induced or topoisomerase DPCs.

In addition, I have some technical and conceptual concerns. However, even if those were addressed, I do not feel that the manuscript provides enough conceptual advancement to warrant publication in The EMBO Journal.

MAJOR CONCERNS

RNF4 depletion by siRNA has a clear effect on the disappearance of chromatin-bound DNMT1 in one experiment (Fig. 1F), but the effect is less obvious in a second similar experiment (S1E). Moreover, the corresponding phenotype in RNF4 KO cells is weak (S1F). This suggests that the contribution of RNF4 to DNMT1 degradation is only minor.

The majority of crosslinked DNMT1 has disappeared in RNF4 KO cells after 3 hours (S1F). However, there is still a strong remaining SUMO signal. siRNA-mediated depletion of RNF4 shows a similar phenotype (S1E). This indicates that other proteins become SUMOylated upon DNMT1 trapping as well, which may suggest that SUMOylation is not DPC-specific. To this reviewer these data indicate that RNF4 is particularly important for degrading these SUMO-targets, but perhaps not the DPC itself.

The authors further suggest that PIAS4 is responsible for DPC SUMOylation. The corresponding data in frog extracts is clear and convincing but the results obtained in cells speak against a major role of PIAS4 in DPC SUMOylation in human cells. Fig S3G shows that there is barely an effect on DNMT1-SUMOylation (apart from a minor difference at 15min).

Many experiments lack crucial controls, which makes their interpretation rather challenging:

- i) Fig 5D only shows mitotic cells treated with SUMOi but does not show untreated cells or non-mitotic cells
- ii) Fig 5D: no control siRNA
- iii) 5G, S4B: no control siRNA
- iv) S4C: only cells treated with SUMOi are shown, no untreated control
- v) Several chromatin fractionation experiments (1F, 1I, S1F, S1E, 4B) lack controls for total (or at least soluble) proteins. It is thus impossible to know how total protein levels are affected. In addition, it would be good to include detection of a soluble, non-chromatin-bound protein as control for the fractionation.
- vi) Several IP-experiments omit crucial blots against input samples (1D, 1E, S2G).

MINOR CONCERNS

If PIAS4 is indeed upstream of RNF4, one would expect that PIAS4 KO cells are sensitive to 5-aza-dC and that this sensitivity is epistatic to RNF4 loss. Is this the case?

The authors claim that "DPCs do not activate interphase damage checkpoints". Is this also the case in non-transformed cells, non-tumour cells?

The effect of Mg262 on the stabilization of the DPC in Fig 3J is not that clear to this reviewer. Perhaps a quantification of independent experiments would help to make the case.

Page 9: Fig. 4B is referenced instead of 3B.

NON-ESSENTIAL SUGGESTIONS

The discussion refers to "data not shown" on page 17. I would suggest to either provide the data or refrain from discussing it.

In this reviewer's opinion, the manuscript would benefit from a more inclusive discussion of the available literature. Several studies describing the involvement of STUBLs in DPC repair, have not been cited (Heideker et al., Plos Gen., 2011, Steinacher et al., Plos One, 2013, Wei et al., Mol. Cell, 2017). Moreover, it might be worth to discuss that the DPC protease Wss1 has been genetically linked to STUBLs in yeast (Mullen et al. 2010, Sharma et al. 2017).

Referee #2:

This manuscript by Liu et al. describes the mechanism of DNA replication-independent DNA-protein crosslink repair. Previously the authors demonstrated that SUMOylation plays an important role in the resolution of DNA-protein crosslinks (DPCs) that are formed after DNA replication (Borgermann et al. 2019). In this manuscript the authors further establish a mechanism in which SUMOylation of a DPC induces ubiquitination of the crosslinked protein by the SUMO-targeted ubiquitin ligase RNF4 followed by its proteasomal degradation. Using the *Xenopus* egg extract system they show that this DPC SUMOylation is independent of DNA replication and requires the SUMO ligase PIAS4. In addition, they demonstrate that DPC SUMOylation does not affect replication-dependent DPC repair. Surprisingly, a nuclear *Xenopus* egg extract does not support ubiquitination of the SUMOylated DPC but this can be induced by addition of recombinant RNF4 or a mitotic egg extract, indicating a similar mechanism as observed in human cells. The authors then show that persistent DPCs on dsDNA lead to a mitotic arrest without inducing a potent DNA damage checkpoint response. This arrests is caused by a delay in chromosome alignment during mitosis causing segregation defects. Consistent with this, RNF4 depleted cells are hypersensitive to 5-azadC treatment. So, while high levels of persistent DPCs fail to activate a checkpoint response their removal does seem to be important for genome stability and cell survival.

This work describes a new, replication-independent, DPC removal mechanism that is important for

genome stability. This is an important finding that will be of interest to many researchers in the genome maintenance and ubiquitin/SUMO fields. The manuscript contains several elegant techniques that are optimally used to address the questions and the experiments are complete and of high quality. To my opinion this manuscript is very suited for publication in EMBO journal but I do have a few issues I would like to see addressed.

1) The use of 5-azadC to induce post-replicative DPCs is not well explained. Although the authors use the same approach in their previous work, it would be make it a lot easier for the reader if this was explained in some more in detail at the start of the result section.

2) In several assays presented knockdown of RNF4 shows a much less dramatic phenotype compared to SUMOi, this suggests a redundant SUMO-dependent mechanism. Have the author tried double depletion of RNF4 and RNF111?

3) The SUMO-Hpall fusion protein is a much better target for poly-SUMOylation compared to Hpall, why is this? And what is the rationale for using the SUMO-Hpall fusion protein? This should be better explained.

4) In general the addition of the first SUMO moiety seems to have very different kinetics from the subsequent poly-SUMOylation. In the Hpall DPC this first step seems to be very slow, while in PARP this seems to be very fast and independent of PARPi or damage. How is this explained. Could there be a different E3 ligase involved? This should be discussed.

5) It is rather puzzling that the SUMOylation of the Hpall DPC occurs in NPE, while the ubiquitination does not. Instead, the ubiquitination takes place in a mitotic egg extract but it is not shown whether the SUMOylation takes place that mitotic extract (I could have missed it but it seems that the mitotic extract is always added after SUMOylation has taken place). This should be clarified. You would expect that both modifications should be able to take place in one type of extract. If the ubiquitination is induced by CDK activation, could this activity be induced in NPE to promote ubiquitination?

6) One relevant question concerns the number of DPCs that are induced in cells upon azadC treatment and SUMOi or RNF4 knockdown. This is relevant for the observed mitotic defect and following cell sensitivity. Would these levels of DPCs ever occur in normal cells? If not, this may explain the lack of DNA damage checkpoint activation. This should be discussed.

Minor points:

- p9, Fig 4B, should be Fig 3B

- On p10, based on Fig S3F, it is implicated that the levels of RNF4 are very low in NPE. From Fig S3F you cannot draw this conclusion. This could reflect the quality of the antibody. Absolute levels can only be derived when a known amount of purified protein is titrated next to it.

- The representation of the cell cycle FACS data in Fig 4 is difficult to read. In my opinion it would be better to show a quantification with error bars.

- I am confused about Fig S3a, there is almost complete block in DPC repair upon UBi. The authors previously shown that there is a redundant Spartan-dependent pathway, is Spartan depleted in this experiment?

Referee #3:

The authors' group previously reported that DNMT1 DPCs induced by 5-azadC undergo SUMOylation and subsequent degradation by the proteasome. In this manuscript, Liu et al. demonstrate that the SUMO-targeted ubiquitin ligase RNF4 is responsible for the ubiquitination of SUMOylated DNMT1 DPCs. The authors then use a *Xenopus* cell-free system to show that a model DPC substrate is SUMOylated by the E3 ligase PIAS4 in a replication-independent manner, and can undergo ubiquitination and proteasomal degradation if RNF4 is supplied. Finally, the authors demonstrate that unresolved DNMT1 DPCs in RNF4-depleted cells cause mitotic defects, highlighting the importance of the SUMO-driven mechanism of DPC repair for maintaining chromosome stability.

This study addresses an important question regarding the replication-independent mechanism of DPC repair. While much progress has been made in understanding the mechanisms of replication-coupled DPC repair, replication-independent mechanisms remain enigmatic. The findings reported in this manuscript provide novel insights into our understanding of how cells deal with DPCs independent of DNA replication and the biological significance of such mechanisms.

Overall, the presented data are high quality and the results were interpreted properly. However, several points listed below need to be addressed to establish more direct evidence for the role of DPC SUMOylation. In addition, while the effect of unresolved DNMT1 DPCs on mitosis is significant in terms of understanding the potential effect of the drug, whether such effect is shared among different DPCs that form after DNA replication has not been addressed sufficiently.

Major points:

(1) More evidence is needed to support their claim that SUMO-dependent DPC repair is indeed mediated by SUMO modifications of DPCs. Current experiments heavily rely on SUMOi, which essentially shuts down all SUMOylation in cells, and that makes it difficult to attribute its effect to SUMOylation of DPCs only. Now that the authors have identified PIAS4 as a SUMO E3 ligase for DPCs, they could strengthen their claim by demonstrating more direct connections between PIAS4 and DNMT1 DPCs. For example, recruitment of PIAS4 to DNMT1 DPCs was not demonstrated. The authors need to examine colocalization of PIAS4 with 5-azadC-induced DNMT1 foci as well as interaction of PIAS4 with DNMT1 after 5-azadC treatment.

(2) Fig. S2GH: The *in vivo* evidence for the role of PIAS4 in DNMT1 SUMOylation is weak. In particular, it is puzzling that siPIAS4 #2 reduced DPC ubiquitination without major effect on DPC modification with SUMO2/3. This cannot be explained by "compensatory activities of other SUMO E3 ligases" (Page 9, line 8) because DPC ubiquitination was impaired by siPIAS4. Rather, it might suggest that PIAS4 plays an indispensable role in this pathway but not through SUMO2/3 modifications. For example, a recent study by Sun et al. showed a stronger effect of PIAS4 depletion on SUMO1 modification of TOP1 and TOP2 DPCs than SUMO2/3 modifications (DOI: 10.1126/sciadv.aba6290). The authors need to clarify the role of PIAS4 in DPC SUMOylation and in promoting ubiquitination by RNF4.

(3) The notion that unresolved DPCs impair mitosis is novel, but it was largely derived from experiments on DNMT1 DPCs. It is important to know whether the effect of DPCs on mitosis is common among various DPCs. The authors briefly addressed this by showing hypersensitivities of RNF4 knock down cells to formaldehyde, but such experiments might not necessarily reflect the role of RNF4 in DPC removal given that formaldehyde causes other types of DNA damage and RNF4 is widely involved in DNA damage response. The role of RNF4 in the clearance of

formaldehyde-induced DPCs needs to be demonstrated, and then the impact of such DPCs on mitosis should be examined by treating RNF4-depleted G2 cells with physiologically relevant concentrations of formaldehyde.

Minor points:

Fig. 2l: The figure indicates U2OS was used but the legend describes HeLa/GFP-DNMT1. If U2OS was used, PIAS4 Western blotting should be shown. In addition, the authors need to indicate which siPIAS4 (#1 or #2) was used in Fig. 2l.

Point-by-point reply to the referees' comments

We would like to thank the referees for the constructive and insightful remarks and suggestions they made on our study. In the revised version of our manuscript, we have included the results of a range of new experiments performed on the basis of the reviewers' helpful comments. In addition, a number of points have been clarified in the text. Collectively, we believe that the new additions to the manuscript address the referees' key concerns and strengthen our original conclusion that DNA replication-independent repair of DNA-protein crosslinks in duplex DNA via the SUMO-RNF4 pathway has a critical role in protecting against chromosomal instability, as explained in the detailed point-by-point response to the referee reports (replicated in full) below.

Referee #1:

GENERAL SUMMARY AND SIGNIFICANCE

The manuscript by Liu et al reports a follow-up study to their recent publication showing that covalent DNA-protein crosslinks undergo SUMOylation-dependent but replication-independent degradation by the proteasome and the metalloprotease ACRC (Borgermann et al EMBO 2019). The authors use cell-based assays and experiments conducted in X. laevis egg extracts to identify the PIAS4 SUMO E3 ligase as being responsible for DPC SUMOylation and the STUBL RNF4 for subsequent ubiquitylation. Similar to what has just been suggested for topoisomerase-DPCs by the Pommier lab (Sun et al., 2020). The authors find that DPC SUMOylation is unrelated to replication-coupled repair of DPCs, but is crucial for the removal of protein adducts prior to mitosis in an RNF4-dependent but replication-independent manner.

The manuscript provides new insights regarding the identity of the enzymes modifying DPCs for repair, but the general concept of SUMO-dependent degradation of DPCs has been proposed previously (Borgermann et al, EMBO 2019, Fielden et al, Nat Comm 2020). The authors analyse the fate of trapped DNMT1 in cells but draw general conclusion regarding the repair of DPCs. Thus, claims in title/abstract should either be reduced to DNMT1-DPCs or the cellular data has to be extended to aldehyde-induced or topoisomerase DPCs.

Our cell-based experiments mainly focused on 5-azadC-induced DNMT1 DPCs, since these lesions can be generated post-replicatively on duplex DNA in a relatively well-defined manner and are not accompanied by DNA breakage, unlike topoisomerase DPCs. We agree with the reviewer that extending the studies of DNMT-type DPCs to other DPCs residing in otherwise undamaged duplex DNA is of key importance for establishing whether the observed mechanisms and impact of DNMT1 DPCs represent general features of this class of DPCs. Accordingly, as suggested by the referee, we performed a series of new experiments to carefully analyze whether the repair mechanisms for DNMT1 DPCs and the impact of blocking their resolution on mitotic progression and genome stability also apply to formaldehyde-induced DPCs. First, we found that exposing cells to a dose of formaldehyde that potently induces DPC formation and an accompanying dramatic chromatin SUMOylation response (Borgermann et al., EMBO J. 38:e101496 (2019)) only has a minor impact on canonical DNA damage signaling (new Fig. EV4F), similar to the effect of 5-azadC-induced DPCs. Second, consistent with the hypersensitivity of RNF4-depleted cells to formaldehyde (Fig. EV5D), we show that RNF4 undergoes strong recruitment to chromatin upon DPC formation by formaldehyde treatment in a SUMOylation-dependent manner and is needed for

efficient resolution of the resulting chromatin-associated SUMO foci (new Fig. EV11,J). Third, using live-cell imaging of cells exposed to formaldehyde treatment in G2 phase to avoid adverse impacts on DNA replication, we show that suppressing the resolution of formaldehyde-induced DPCs via SUMOi treatment delays progression through mitosis and greatly enhances the incidence of defective chromosome segregation (new Fig. EV5B,C), again paralleling the response to 5-azadC-induced DPCs. Collectively, these new data show that RNF4 is also involved in processing formaldehyde-induced DPCs and that these lesions do not trigger G2 cell cycle arrest but undermine mitotic fidelity similar to DNMT1 DPCs if left unresolved, suggesting that these features apply generally to DPCs in uninterrupted duplex DNA. Together with a recent study from the Pommier laboratory referred to by the reviewer, our findings thus indicate that RNF4 has a general role in processing SUMOylated DPCs, including both DPCs in uninterrupted duplex DNA (this study) and DPCs flanked by DNA breaks (TOP1 and TOP2 DPCs; Sun et al., Sci Adv 6:eaba6290 (2020)).

In addition, I have some technical and conceptual concerns. However, even if those were addressed, I do not feel that the manuscript provides enough conceptual advancement to warrant publication in The EMBO Journal.

We respectfully disagree that our manuscript does not provide sufficient conceptual advancement, as relatively little is known about how DPCs are resolved by DNA replication-independent mechanisms, particularly in the case of DPCs residing in uninterrupted duplex DNA. Although the general concept of SUMO-dependent degradation of DPCs has been established in recent work by others and us as pointed out by the reviewer above, mechanistically how SUMOylation drives the removal of post-replicative DPCs, its interrelation with DNA replication-coupled DPC repair mechanisms and biological significance in preventing genomic instability have remained central yet unresolved issues. We believe our study addresses these key questions and thus provides an important advancement of the field.

MAJOR CONCERNS

RNF4 depletion by siRNA has a clear effect on the disappearance of chromatin-bound DNMT1 in one experiment (Fig. 1F), but the effect is less obvious in a second similar experiment (S1E). Moreover, the corresponding phenotype in RNF4 KO cells is weak (S1F). This suggests that the contribution of RNF4 to DNMT1 degradation is only minor.

While RNF4 knockdown or knockout reduces the kinetics of DNMT1 DPC resolution in U2OS, HeLa and HAP1 cells, an important conclusion arising from our work is that loss of RNF4 only partially impairs DPC removal unlike the effect of SUMOi. This clearly suggests that additional SUMO-mediated but RNF4-independent mechanisms for DPC resolution exist, a notion we emphasize in the manuscript as follows: “We note, however, that unlike complete inhibition of DNMT1 DPC degradation by SUMOi, loss of RNF4 only partially suppressed the removal of SUMO-modified DNMT1 DPCs (Fig. 1G; Fig. EV1E,F), suggesting the existence of additional, SUMO-driven but RNF4-independent mechanisms for post-replicative DNMT1 DPC resolution.” (page 8). Importantly, however, even though loss of RNF4 only delays but does not abolish the clearance of DPCs, this is sufficient to induce chromosome segregation defects and hypersensitivity to DPC formation (Fig. 5C,D,F,G). The existence of both RNF4-dependent and -independent mechanisms for processing SUMO-modified DPCs could help to ensure efficient clearance of these

lesions prior to mitotic entry, thereby mitigating the risk of mitotic chromosome segregation errors, and the relative usage of individual pathways may vary between cell types. The nature of SUMO-dependent but RNF4-independent DPC resolution mechanisms remains to be established but is a topic of ongoing investigation in our labs. For instance, our preliminary evidence suggests a potential role of the transcriptional machinery in RNF4-independent but SUMO-dependent clearance of DNMT1 DPCs in human cells (Fig. R1), although the precise mechanistic basis of this observation is not yet clear and goes beyond the scope of the current manuscript.

See also the point below.

Fig. R1.

A potential role of the transcription machinery in SUMO-dependent DPC resolution

Combined RNF4 knockdown and transcriptional inhibition impairs DNMT1 DPC resolution to a similar extent as SUMOi treatment. U2OS cells transfected with non-targeting control (CTRL) or RNF4 siRNA were synchronized in early S phase by release from double thymidine block. Cells were then pre-treated with the transcription inhibitor DRB or SUMOi for 15 min where indicated, exposed to 5-azadC for 30 min and collected at the indicated times after 5-azadC withdrawal. DNMT1 foci enumeration was performed using quantitative image-based cytometry (>8700 cells analyzed per condition). Data from a representative experiment are shown.

The majority of crosslinked DNMT1 has disappeared in RNF4 KO cells after 3 hours (S1F). However, there is still a strong remaining SUMO signal. siRNA-mediated depletion of RNF4 shows a similar phenotype (S1E). This indicates that other proteins become SUMOylated upon DNMT1 trapping as well, which may suggest that SUMOylation is not DPC-specific. To this reviewer these data indicate that RNF4 is particularly important for degrading these SUMO-targets, but perhaps not the DPC itself.

We thank the reviewer for raising this important point. While we demonstrate that SUMOylation of DNMT1 DPCs promote their subsequent RNF4-mediated modification by K48-linked polyubiquitylation and proteasome-dependent removal in cells, it is indeed possible that RNF4 targets additional SUMOylated proteins at DPC sites. We previously showed using quantitative mass spectrometry that 5-azadC-induced DPCs trigger a highly targeted SUMOylation response that predominantly impinges on DNMT1 itself (Borgermann et al., EMBO J. 38:e101496 (2019)). However, a small subset of additional proteins, several of which are known DNMT1-interacting proteins (e.g PCNA and UHRF1), also show increased SUMOylation under these conditions, in a manner that is fully dependent on DNMT1 DPC formation (Fig. 2 in Borgermann et al., EMBO J. 38:e101496 (2019)). The 5-azadC-induced chromatin SUMOylation response thus appears to conform well to the established ‘SUMO group modification’ principle, in that it targets both crosslinked DNMT1 molecules and, to a lesser extent, proteins residing in proximity to the DPC. In at least some experiments (e.g. Fig. EV1E,F), as pointed out by the reviewer, we see a trend that overall SUMO modifications take longer to clear from chromatin following 5-azadC treatment than the trapped DNMT1 molecules themselves, regardless of whether RNF4 is present or not. Possible reasons for this apparent uncoupling include that the SUMO2/3 antibody used in our experiments gives a much stronger signal in immunoblots than the DNMT1 antibody, and/or that some epitopes recognized by the DNMT1 antibody may become masked due to post-translational modifications of DNMT1 following its covalent trapping on DNA. However, in line with the SUMO group modification principle, we consider it likely that RNF4 does not exclusively target the adducted protein but may also promote the ubiquitylation and turnover of other SUMOylated proteins at DPC sites as part of re-establishing an intact chromatin state following lesion removal. To highlight this possibility, we added the following statement to the revised manuscript: “*We previously showed that while DNMT1 is the main cellular target of 5-azadC-induced SUMOylation, a small range of additional proteins including known DNMT1-binding factors also display increased SUMOylation upon DNMT1 DPC formation (Borgermann et al., 2019). Because loss of RNF4 led to a marked delay in reversing overall 5-azadC-induced chromatin SUMOylation (Fig. 1F; Fig. EV1E,G), it is likely that RNF4 STUbL activity is not exclusively targeted to DPCs but also impacts other SUMOylated proteins at DPC sites to facilitate lesion removal and re-establishment of a normal chromatin state.*” (page 7). It should be emphasized, however, that we also show direct RNF4-dependent ubiquitylation of SUMOylated DNMT1 DPCs *in vitro* and SUMOylated M.HpaII DPCs in *Xenopus* egg extracts (Fig. 1J; Fig. 3E,J). Thus, RNF4 clearly targets the DPC itself and the main conclusion of our manuscript remains unchanged.

The authors further suggest that PIAS4 is responsible for DPC SUMOylation. The corresponding data in frog extracts is clear and convincing but the results obtained in cells speak against a major role of PIAS4 in DPC SUMOylation in human cells. Fig S3G shows that there is barely an effect on DNMT1-SUMOylation (apart from a minor difference at 15min).

The reviewer is correct in pointing out that whereas in *Xenopus* egg extracts PIAS4 is instrumental for SUMOylation of single defined M.HpaII DPCs located in a plasmid context and directly SUMOylates these DPCs *in vitro* (Fig. 2), it is largely dispensable for SUMOylation of DNMT1 DPCs distributed across the genome in human cells. To better characterize the role of PIAS4 in human cells, we now show that PIAS4 is recruited to DNMT1 DPC sites and shows increased binding to DNMT1 upon DPC formation (new Fig. 2I,J), suggesting its involvement in promoting SUMOylation and/or resolution of these DPCs. However, despite its clear recruitment, depletion of PIAS4 had no overt impact on DNMT1 DPC SUMOylation (new Fig. EV2K,L). In fact, we found that individual depletion of other established SUMO E3 ligases did not impair overall 5-azadC-

induced DNMT1 SUMOylation (new Fig. 2K). Together, these observations suggest that while PIAS4 is likely involved in the response to DNMT1 DPCs in human cells, there is considerable redundancy between individual SUMO E3s in driving DNMT1 DPC SUMOylation. The observed discrepancy between the relative importance of PIAS4 for DPC SUMOylation in *Xenopus* egg extracts and human cells may at least partially reflect the distinct DPC contexts analyzed in the two systems (i.e. plasmid in *Xenopus* egg extracts vs. chromatin in human cells), and/or the absence in egg extracts of active transcription, which in cells may potentially provide an additional DPC-sensing mechanism. The redundancy between different SUMO E3s in promoting DNMT1 DPC SUMOylation in cells seems well in line with the overall strong magnitude of this response.

Despite the greater complexity of DPC SUMOylation in cells, which we point out and discuss in the revised manuscript (pages 10 and 19), the clear ability of PIAS4 to SUMOylate DPCs in egg extracts and *in vitro* supports and extends recent data from the Pommier laboratory (Sun et al., Sci Adv 6:eaba6290 (2020)) describing an involvement of PIAS4 in TOP1/2 DPC repair, thus highlighting a conserved role for PIAS4 in the resolution of different types of DPCs. Although further work is needed to clarify individual contributions of different SUMO E3 ligases to DNMT1 DPC SUMOylation in cells, we feel that the ability of PIAS4 to SUMOylate DPCs on duplex DNA is an important finding to report, and we therefore opted to keep this data in the new version of the manuscript.

Many experiments lack crucial controls, which makes their interpretation rather challenging:
i) Fig 5D only shows mitotic cells treated with SUMOi but does not show untreated cells or non-mitotic cells

We believe the reviewer may be referring to Fig. 4B in the original manuscript. Considering also the referee's point (v) below, we have replaced this data with a new experiment, which includes both soluble and chromatin fractions from cells treated or not with SUMOi (new Fig. 4B).

ii) Fig 5D: no control siRNA

We assume the reviewer is referring to the representative images that were originally shown in Fig. 4D. We removed this panel from the revised manuscript, but data quantification for both siCTRL- and siRNF4-treated cells is shown in Fig. 4C.

iii) 5G, S4B: no control siRNA

We have added the missing control experiments (new Fig. 4F; Fig. EV4B-D).

iv) S4C: only cells treated with SUMOi are shown, no untreated control

We have added the corresponding data for untreated RPE-1 cells (new Fig. EV4B).

v) Several chromatin fractionation experiments (1F, 1I, S1F, S1E, 4B) lack controls for total (or at least soluble) proteins. It is thus impossible to know how total protein levels are affected. In addition, it would be good to include detection of a soluble, non-chromatin-bound protein as control for the fractionation.

We have added blots of soluble fractions (incl. NF- κ B p65, a soluble, non-chromatin-bound protein) as controls in the cell fractionation experiments (Fig. 1F,I; Fig. EV1E,G,I; Fig. 4B).

vi) Several IP-experiments omit crucial blots against input samples (1D, 1E, S2G).

Input samples for the IP experiments in the manuscript (Fig. 1D,E; Fig. 2J,K; Fig. EV2K) have now been added.

We apologize for the previous omission of these controls.

MINOR CONCERNS

If PIAS4 is indeed upstream of RNF4, one would expect that PIAS4 KO cells are sensitive to 5-aza-dC and that this sensitivity is epistatic to RNF4 loss. Is this the case?

We tested this and found that PIAS4 depletion mildly enhances cellular sensitivity to 5-azadC (Fig. R2). Moreover, PIAS4 and RNF4 loss appear largely epistatic in sensitizing cells to 5-azadC (Fig. R2). This suggests that PIAS4 and RNF4 may act in a joint pathway for DNMT1 DPC resolution in cells, as observed in *Xenopus* egg extracts. However, while PIAS4 is clearly recruited to DNMT1 DPCs in cells, its depletion does not overtly impair DNMT1 DPC SUMOylation as discussed above. Thus, the underlying reason for the mild sensitization of cells depleted of PIAS4 to 5-azadC treatment is currently unclear, and we therefore prefer not to include these clonogenic survival data in the revised manuscript.

Fig. R2.

Impact of PIAS4 and RNF4 knockdown on cellular sensitivity to 5-azadC

Clonogenic survival of 5-azadC-treated HeLa cells transfected with indicated siRNAs (mean \pm SEM; $n=2$ independent experiments).

The authors claim that "DPCs do not activate interphase damage checkpoints". Is this also the case in non-transformed cells, non-tumour cells?

Yes – we find that despite triggering DPC formation and SUMOylation, 5-azadC treatment of non-transformed, non-tumor RPE-1 cells does not induce a detectable level of conventional DNA damage signaling (Fig. EV4E), suggesting that DPCs residing in uninterrupted duplex DPCs have a negligible impact on activating canonical interphase DNA damage checkpoints.

The effect of Mg262 on the stabilization of the DPC in Fig 3J is not that clear to this reviewer. Perhaps a quantification of independent experiments would help to make the case.

To better address the role of the proteasome in the SUMO-RNF4-driven DPC resolution pathway, we immunodepleted the proteasome from extracts using an antibody targeting its PSMA1 subunit. As shown in new Fig. 3L, PSMA1 immunodepletion delayed DPC degradation similar to MG262 treatment and also induced longer ubiquitin/SUMO chains on the DPC, further suggesting a role for the proteasome in DPC removal. However, as noted with proteasome inhibitors, the DPC is still degraded in the absence of the proteasome, suggesting that another protease also degrades poly-ubiquitylated DPCs (as also observed in (Larsen et al., Mol Cell 73:574-588 (2019)) during replication-coupled DPC repair).

We have quantified these gels as suggested by the reviewer. However, quantification of M.HpaII consistently shows an increase in signal at the onset of DPC ubiquitylation at the 30 min time point of the reaction, which is likely caused by the non-linearity of ECL (Fig. R3A,B). We therefore prefer not to show quantifications of these blots. Instead, the effect of PSMA1 depletion is now shown in three biological replicates in the revised manuscript (new Fig. 3K,L; Fig. EV3L,M) in addition to the effect of MG262 (Fig. EV3K).

Fig. R3.

Impact of PSMA1 depletion on M.HpaII degradation

(A) p4xDPC^{SUMO} was polySUMOylated in NPE, recovered via plasmid pull-down and incubated in fresh CSF-arrested extract that was either mock- or PSMA1-depleted. At indicated time points following CSF extract addition, the plasmid was recovered and immunoblotted against M.HpaII. This panel is a duplication of Fig. 3L. (B) Quantification of data in (A), illustrating the non-linearity of the M.HpaII signal.

Page 9: Fig. 4B is referenced instead of 3B.

This error has now been corrected.

NON-ESSENTIAL SUGGESTIONS

The discussion refers to "data not shown" on page 17. I would suggest to either provide the data

or refrain from discussing it.

We have removed the statement referring to the data not shown.

In this reviewer's opinion, the manuscript would benefit from a more inclusive discussion of the available literature. Several studies describing the involvement of STUBLs in DPC repair, have not been cited (Heideker et al., Plos Gen., 2011, Steinacher et al., Plos One, 2013, Wei et al., Mol. Cell, 2017). Moreover, it might be worth to discuss that the DPC protease Wss1 has been genetically linked to STUBLs in yeast (Mullen et al. 2010, Sharma et al. 2017).

We thank the reviewer for the useful suggestions. We now mention and cite most of these references (Heideker et al., PLoS Gen., 2011; Steinacher et al., PLoS One, 2013; Wei et al., Mol Cell, 2017; Sharma et al., Genetics 2017) in the introduction and discussion sections. However, we opted not to include the Mullen et al. 2010 study, as the interplay between Wss1 and STUBLs described in that manuscript is in our view more difficult to rationalize and would require a longer discussion that would disrupt the flow of the manuscript.

Referee #2:

This manuscript by Liu et al. describes the mechanism of DNA replication-independent DNA-protein crosslink repair. Previously the authors demonstrated that SUMOylation plays an important role in the resolution of DNA-protein crosslinks (DPCs) that are formed after DNA replication (Borgermann et al. 2019). In this manuscript the authors further establish a mechanism in which SUMOylation of a DPC induces ubiquitination of the crosslinked protein by the SUMO-targeted ubiquitin ligase RNF4 followed by its proteasomal degradation. Using the Xenopus egg extract system they show that this DPC SUMOylation is independent of DNA replication and requires the SUMO ligase PIAS4. In addition, they demonstrate that DPC SUMOylation does not affect replication-dependent DPC repair. Surprisingly, a nuclear Xenopus egg extract does not support ubiquitination of the SUMOylated DPC but this can be induced by addition of recombinant RNF4 or a mitotic egg extract, indicating a similar mechanism as observed in human cells. The authors then show that persistent DPCs on dsDNA lead to a mitotic arrest without inducing a potent DNA damage checkpoint response. This arrest is caused by a delay in chromosome alignment during mitosis causing segregation defects. Consistent with this, RNF4 depleted cells are hypersensitive to 5-azadC treatment. So, while high levels of persistent DPCs fail to activate a checkpoint response their removal does seem to be important for genome stability and cell survival.

This work describes a new, replication-independent, DPC removal mechanism that is important for genome stability. This is an important finding that will be of interest to many researchers in the genome maintenance and ubiquitin/SUMO fields. The manuscript contains several elegant techniques that are optimally used to address the questions and the experiments are complete and of high quality. To my opinion this manuscript is very suited for publication in EMBO journal but I do have a few issues I would like to see addressed.

1) The use of 5-azadC to induce post-replicative DPCs is not well explained. Although the authors use the same approach in their previous work, it would be make it a lot easier for the

reader if this was explained in some more in detail at the start of the result section.

We concur with this notion and now explain more carefully how 5-azadC triggers DNMT1 DPC formation at the start of the Results section, as follows: “*To understand the mechanistic basis of how SUMOylation promotes removal of DPCs, we first monitored the resolution kinetics for DNMT1 DPCs that are formed in the wake of the replication fork following 5-azadC incorporation into genomic DNA during replication. To this aim, cells were synchronized in early S phase, exposed to a brief (30-min) pulse of 5-azadC in order to ensure even incorporation of this nucleotide analog into genomic DNA of all cells, and analyzed at different time points (Fig. 1A). Methylation of 5-azadC by DNMT-type methyltransferases, in particular DNMT1 that methylates CpG motifs in newly replicated DNA to maintain DNA methylation patterns, leads to their covalent crosslinking to DNA.*” (page 6). We hope this makes it clearer to readers how 5-azadC induces post-replicative DPCs.

2) In several assays presented knockdown of RNF4 shows a much less dramatic phenotype compared to SUMO1, this suggests a redundant SUMO-dependent mechanism. Have the author tried double depletion of RNF4 and RNF111?

This is a good suggestion, which we tested. Consistent with a lack of impact of RNF111 knockdown on the ubiquitylation and resolution of SUMO-modified DNMT1 DPCs (Fig. 1E,F), we found that co-depletion of RNF4 and RNF111 does not further impair DNMT1 DPC clearance relative to RNF4 knockdown alone (new Fig. EV1F). This suggests that RNF111 is not responsible for residual SUMO-mediated DNMT1 DPC resolution in RNF4-depleted cells and that other pathways must therefore exist, a notion that we highlight in the revised manuscript (page 7-8). We are currently investigating the molecular underpinnings of such mechanisms.

3) The SUMO-HpaII fusion protein is a much better target for poly-SUMOylation compared to HpaII, why is this? And what is the rationale for using the SUMO-HpaII fusion protein? This should be better explained.

As pointed out by the reviewer, SUMO-M.HpaII is a much better poly-SUMOylation target than M.HpaII (Fig. 2B), but the reason for this is still unclear. We note, however, that PIAS4 contains two SUMO-interacting motifs (SIMs) in its C-terminus, in contrast to other PIAS/SIZ family members that harbor only one (Kaur et al., JBC 292:10230-10238 (2017)). SIM domains in PIAS-type SUMO E3 ligases are thought to be important for interacting with SUMO-conjugated UBC9, the SUMO E2 enzyme, positioning it for catalysis. Thus, it is possible that PIAS4 utilizes its SIMs to interact simultaneously with two different SUMO molecules, one conjugated to UBC9 and one on the substrate, to promote poly-SUMOylation. Such a mechanism would mean that the first SUMO on the DPC is rate-limiting but would then stimulate DPC poly-SUMOylation. Supporting this notion, our mass spectrometry data in Fig. 2E shows that recruitment of PIAS4 to damaged chromatin is SUMOylation-dependent, suggesting that SUMOylated substrates on chromatin stimulate PIAS4 recruitment or retention to DNA to promote further poly-SUMOylation. In agreement with this, we found that in cells PIAS4 is also recruited to DNMT1 DPCs in a SUMOylation-dependent manner (new Fig. 2I).

To begin to address this possibility and understand the interplay between the PIAS4 SIMs and DPC SUMOylation we generated a PIAS4 double SIM mutant, which did not rescue DPC SUMOylation (new Fig. EV2H-J). We are now in the process of addressing the roles of the individual PIAS4

SIMs. However, understanding their precise mechanistic involvements in DPC SUMOylation would likely require substantial further experimental and structural work that we feel go well beyond the scope of the present study.

To better explain the use of the SUMO-M.HpaII DPC plasmid, we added the following sentence to the legend for Fig. 2B in the revised manuscript: “*Note that priming M.HpaII with SUMO stimulates rapid poly-SUMOylation of the DPC in NPE. Thus, this substrate was frequently used in subsequent experiments to stimulate DPC poly-SUMOylation.*”

4) In general the addition of the first SUMO moiety seems to have very different kinetics from the subsequent poly-SUMOylation. In the HpaII DPC this first step seems to be very slow, while in PARP this seems to be very fast and independent of PARPi or damage. How is this explained. Could there be a different E3 ligase involved? This should be discussed.

The reviewer raised the interesting possibility that conjugation of the first SUMO moiety to PARP1 might be driven by a SUMO ligase other than PIAS4 in the absence of DNA damage. We addressed this by incubating untreated sperm chromatin in either mock- or PIAS4-depleted extracts. As can be seen in new Fig. EV2G, PIAS4 depletion abolished PARP1 mono-SUMOylation, which could be rescued by reintroducing PIAS4. This indicates that PIAS4 is responsible for both mono- and poly-SUMOylation of PARP1, as also observed for M.HpaII. Why PARP1 is mono-SUMOylated faster than M.HpaII is unknown but could be explained by the large amount of PARP1 binding to undamaged sperm chromatin (likely because of a high abundance of nicks present in the genome).

5) It is rather puzzling that the SUMOylation of the HpaII DPC occurs in NPE, while the ubiquitination does not. Instead, the ubiquitination takes place in a mitotic egg extract but it is not shown whether the SUMOylation takes place that mitotic extract (I could have missed it but it seems that the mitotic extract is always added after SUMOylation has taken place). This should be clarified. You would expect that both modifications should be able to take place in one type of extract. If the ubiquitination is induced by CDK activation, could this activity be induced in NPE to promote ubiquitination?

SUMOylation and ubiquitylation processes are dynamically controlled by the balance of ubiquitin/SUMO E3s and proteases. NPE is indeed more proficient for M.HpaII DPC SUMOylation than mitotic egg extracts; this is now clarified in the text, and a figure showing this has been added to the revised manuscript (new Fig. EV3G). Why mitotic extracts are less efficient in SUMOylating M.HpaII could at least in part be explained by the lower amount of PIAS4 present in these extracts compared to NPE (new Fig. EV3H) and/or higher SUMO protease activity. Importantly, the whole process of DPC SUMOylation/ubiquitylation can be recapitulated by supplementing mitotic extracts with exogenous PIAS4 (new Fig. EV3I).

The reviewer also raised the interesting possibility that supplementing CDK activity to NPE could induce RNF4-mediated ubiquitylation of the SUMO-modified DPCs. We tested this by adding exogenous Cyclin A1 to NPE to mimic a mitotic state (Moreno et al., Life Sci Alliance, 2:e201900390 (2019); Strausfeld et al., J Cell Science 109:1555–1563 (1996)). While some extracts were clearly responsive and showed increased RNF4-dependent DPC ubiquitylation in the presence of exogenous Cyclin A1 (Fig. R4A,B), we had difficulties to consistently recapitulate these results across different extract batches. The reason for these discrepancies is currently unknown but could reflect the limiting amounts of RNF4 present in NPE (low nM concentrations; new Fig. EV3E),

which might fluctuate between different extract batches. We have therefore not included this data in the manuscript. Importantly, our experiments with synchronized cells clearly show that the whole process of SUMO-targeted ubiquitylation and degradation is highly efficient in interphase (e.g. Fig. 1A). Thus, although CDK activity may potentiate RNF4-mediated DPC ubiquitylation, reaching mitosis is clearly not a prerequisite for this process to occur.

Fig. R4.

Stimulation of RNF4-dependent DPC ubiquitylation by elevated CDK activity

(A) p4xDPC^{SUMO} was polySUMOylated in NPE, recovered via plasmid pull-down and incubated in fresh NPE that was either left untreated or supplemented with Cyclin A1 (200 ng/μL). At indicated time points following CSF extract addition, the plasmid was recovered and immunoblotted against M.HpaII. (B) p4xDPC^{SUMO} was polySUMOylated in NPE, recovered via DPC pull-down and incubated in fresh NPE that was either mock- or RNF4-depleted and supplemented with Cyclin A1 where indicated. Note the RNF4-dependent ubiquitylation observed in the presence of Cyclin A1 (compare lanes 7 and 13).

6) One relevant question concerns the number of DPCs that are induced in cells upon azadC treatment and SUMOi or RNF4 knockdown. This is relevant for the observed mitotic defect and following cell sensitivity. Would these levels of DPCs ever occur in normal cells? If not, this may explain the lack of DNA damage checkpoint activation. This should be discussed.

This is an important point. To address this question, we performed a range of experiments with formaldehyde, a potent DPC inducer, which is present at relatively high intracellular concentrations in humans (approx. 400 μM; European Food Safety Authority (EFSA) Journal 12:3550 (2014)). When inducing DPC formation by exposure of cells to near-physiological concentrations of formaldehyde, which triggers a chromatin SUMOylation response that is similar in overall magnitude to that observed for 5-azadC treatment (Borgermann et al., EMBO J. 38:e101496 (2019)) and thus may be assumed to give rise to a comparable level of DPC formation, we found that SUMOi treatment impaired mitotic progression and chromosome segregation fidelity similar to the impact of DNMT1 DPCs (new Fig. EV5B,C). These findings suggest that the number of DPCs induced under our experimental conditions, which strongly undermine faithful mitotic chromosome segregation when their resolution is blocked (Fig. 5C,D; Fig. EV5C), could in principle be encountered by normal cells, underscoring the importance of SUMO-driven mechanisms for promoting timely DPC removal. We now mention this in the manuscript (page 17-18).

Minor points:

- p9, Fig 4B, should be Fig 3B

This error has now been corrected.

- On p10, based on Fig S3F, it is implicated that the levels of RNF4 are very low in NPE. From Fig S3F you cannot draw this conclusion. This could reflect the quality of the antibody. Absolute levels can only be derived when a known amount of purified protein is titrated next to it.

The levels of RNF4 in NPE and mitotic extracts are now compared to a titration of recombinant RNF4, as suggested by the reviewer (new Fig. EV3E). From this we conclude that RNF4 is present at a low nM concentration in egg extracts.

- The representation of the cell cycle FACS data in Fig 4 is difficult to read. In my opinion it would be better to show a quantification with error bars.

We believe the current standard representation of the FACS data in Fig. 4 and Fig. EV4 is a useful way to display full cell cycle profiles. However, to facilitate the readers' interpretation of the data, we have now indicated for all individual FACS profiles the percentage of cells in G2/M phase.

- I am confused about Fig S3a, there is almost complete block in DPC repair upon UBi. The authors previously shown that there is a redundant Spartan-dependent pathway, is Spartan depleted in this experiment?

SPRTN was not depleted in the experiments shown in Fig. EV3A,B. As pointed out by the reviewer, we previously showed that a redundant SPRTN pathway also acts on a methylated DPC that cannot be ubiquitylated, from which we concluded that ubiquitylation of DPCs is dispensable for their SPRTN-dependent, replication-coupled proteolysis (Larsen et al. 2019). However, in the same study we also showed that SPRTN-mediated DPC proteolysis is dependent on SPRTN's ubiquitin-binding UBZ domain (Fig. 4E and S4G-H in Larsen et al. 2019). Thus, SPRTN function likely depends on the ubiquitylation of a protein other than the DPC itself (possibly PCNA). Consistently, we show below that addition of the ubiquitin-E1 inhibitor (UBi) abolishes SPRTN-mediated degradation of methylated M.HpaII DPCs (Fig. R5A,B). However, we feel that including this data would break the flow of the manuscript, which focuses on replication-independent DPC repair-mechanisms. Instead, we added a specification in the legend for Fig. EV3B to clarify that both the proteasome and SPRTN are inhibited by UBi treatment.

Fig. R5.**Inhibition of SPRTN-mediated DPC proteolysis by UBi**

(A) A plasmid containing two methylated M.HpaII DPCs (pme-DPC^{2xLead}; Larsen et al., Mol Cell 2019) was replicated in egg extracts in the presence of radiolabeled nucleotides and, where indicated, MLN-7243 (UBi; 133 μ M). Samples were analyzed by native agarose gel electrophoresis. Note the absence of supercoiled (SC) repair products in the presence of UBi. (B) Samples from (A) were recovered at the indicated time points via DPC pull-down and blotted against M.HpaII. Note that methylated M.HpaII undergoes replication-coupled proteolysis by SPRTN, which can be visualized by the disappearance of full-length M.HpaII and appearance of degradation fragment (lanes 1-4). In the presence of UBi, full-length methylated M.HpaII is stabilized, confirming inhibition of SPRTN-mediated proteolysis.

Referee #3:

The authors' group previously reported that DNMT1 DPCs induced by 5-azadC undergo SUMOylation and subsequent degradation by the proteasome. In this manuscript, Liu et al. demonstrate that the SUMO-targeted ubiquitin ligase RNF4 is responsible for the ubiquitination of SUMOylated DNMT1 DPCs. The authors then use a Xenopus cell-free system to show that a model DPC substrate is SUMOylated by the E3 ligase PIAS4 in a replication-independent manner, and can undergo ubiquitination and proteasomal degradation if RNF4 is supplied. Finally, the authors demonstrate that unresolved DNMT1 DPCs in RNF4-depleted cells cause mitotic defects, highlighting the importance of the SUMO-driven mechanism of DPC repair for maintaining chromosome stability.

This study addresses an important question regarding the replication-independent mechanism of DPC repair. While much progress has been made in understanding the mechanisms of replication-coupled DPC repair, replication-independent mechanisms remain enigmatic. The findings reported in this manuscript provide novel insights into our understanding of how cells deal with DPCs independent of DNA replication and the biological significance of such mechanisms.

Overall, the presented data are high quality and the results were interpreted properly. However, several points listed below need to be addressed to establish more direct evidence for the role of DPC SUMOylation. In addition, while the effect of unresolved DNMT1 DPCs on mitosis is significant in terms of understanding the potential effect of the drug, whether such effect is shared among different DPCs that form after DNA replication has not been addressed sufficiently.

Major points:

(1) More evidence is needed to support their claim that SUMO-dependent DPC repair is indeed mediated by SUMO modifications of DPCs. Current experiments heavily rely on SUMOi, which essentially shuts down all SUMOylation in cells, and that makes it difficult to attribute its effect to SUMOylation of DPCs only. Now that the authors have identified PIAS4 as a SUMO E3 ligase for DPCs, they could strengthen their claim by demonstrating more direct connections between PIAS4 and DNMT1 DPCs. For example, recruitment of PIAS4 to DNMT1 DPCs was not demonstrated. The authors need to examine colocalization of PIAS4 with 5-azadC-induced DNMT1 foci as well as interaction of PIAS4 with DNMT1 after 5-azadC treatment.

Following the reviewer's suggestion, we analyzed whether PIAS4 is recruited to DNMT1 DPCs. In the revised manuscript, we included new data showing that PIAS4 colocalizes with 5-azadC-induced DNMT1 foci (new Fig. 2I). Moreover, we show that PIAS4 displays increased interaction with DNMT1 after 5-azadC treatment (new Fig. 2J), suggesting its involvement in promoting DNMT1 DPC SUMOylation (please see also our response to point 2 below).

Whether SUMO-dependent DPC repair is mediated by direct SUMOylation of the DPC is an important question. To this end, we provide evidence that SUMOylated DNMT1 immunopurified from cells is a direct substrate for the STUbL activity of RNF4 *in vitro* (Fig. 1J), and that upon DPC formation DNMT1 undergoes robust RNF4-dependent polyubiquitylation via K48 linkages (Fig. 1D,E), a key signal for degradation via proteasomal activity, which is required for DNMT1 DPC resolution (Fig. 1C). Likewise, we demonstrate RNF4-dependent ubiquitylation of SUMOylated M.HpaII DPCs in *Xenopus* egg extracts (Fig. 3E,J). Together, this strongly suggests that direct SUMOylation of DPCs trigger their subsequent polyubiquitylation and degradation. However, a formal demonstration that SUMOylation of DPCs *per se* mediates their ubiquitin-dependent modification and resolution in cells would require the generation of a DNMT1 mutant that is refractory to 5-azadC-induced polySUMOylation. Our mass spectrometry data indicates that DNMT1 undergoes SUMOylation on numerous lysine residues in response to its 5-azadC-induced trapping on DNA (our unpublished observations), suggesting a high degree of promiscuity at site level in the SUMOylation of DPCs, which would be expected for an efficient mechanism that nonspecifically primes adducted proteins for proteolytic destruction. Given this, we do not think it is feasible to generate a DNMT1 mutant that is resistant to SUMOylation upon DPC formation while retaining functionality. Thus, although we demonstrate that SUMOylated DPCs are targeted by RNF4 for K48-linked polyubiquitylation that provides a signal for proteasomal degradation, we cannot presently rule out that SUMOylation of other proteins also contributes to productive DPC repair. In the case of DNMT1 DPCs, we previously showed using quantitative mass spectrometry that 5-azadC-induced DPCs trigger a remarkably targeted SUMOylation response that predominantly impinges on DNMT1 itself (Borgermann et al., EMBO J. 38:e101496 (2019)). However, a small subset of additional proteins, several of which are known DNMT1-interacting proteins (e.g UHRF1), also show increased SUMOylation under these conditions albeit to a much lesser extent, and in a manner that is fully dependent on DNMT1 DPC formation (Fig. 2 in Borgermann et al., EMBO J. 38:e101496 (2019)). Accordingly, SUMOylation of these proteins at DPC sites might also contribute to facilitating efficient lesion removal and re-establishment of a normal chromatin state. We now mention this possibility in the revised manuscript (page 7).

(2) Fig. S2GH: The *in vivo* evidence for the role of PIAS4 in DNMT1 SUMOylation is weak. In particular, it is puzzling that siPIAS4 #2 reduced DPC ubiquitination without major effect on DPC modification with SUMO2/3. This cannot be explained by "compensatory activities of other SUMO E3 ligases" (Page 9, line 8) because DPC ubiquitination was impaired by siPIAS4. Rather, it might suggest that PIAS4 plays an indispensable role in this pathway but not through SUMO2/3 modifications. For example, a recent study by Sun et al. showed a stronger effect of PIAS4 depletion on SUMO1 modification of TOP1 and TOP2 DPCs than SUMO2/3 modifications (DOI: 10.1126/sciadv.aba6290). The authors need to clarify the role of PIAS4 in DPC SUMOylation and in promoting ubiquitination by RNF4.

We concur with this notion and thank the reviewer for the constructive suggestions. Prompted by these, we performed a range of additional experiments to carefully assess the impact of a range of independent PIAS4 siRNAs on 5-azadC-induced DNMT1 SUMOylation and ubiquitylation. While

PIAS4 is recruited to DNMT1 DPC sites and displays 5-azadC-stimulated interaction with DNMT1 (new Fig. 2I,J), the overall conclusion emerging from our PIAS4 knockdown studies is that this has no overt impact on DNMT1 SUMO modifications via both SUMO2/3 and SUMO1 (new Fig. EV2K,L). In fact, individual depletion of other established SUMO E3 ligases also did not impair overall 5-azadC-induced DNMT1 SUMOylation (new Fig. 2K). Together, these observations suggest that while PIAS4 is clearly targeted to DNMT1 DPCs in human cells, there is redundancy between individual SUMO E3s in SUMOylating these DPCs. We consider it likely that the observed discrepancy between the relative importance of PIAS4 for DPC SUMOylation in *Xenopus* egg extracts and human cells may at least partially reflect the distinct nature of the DPCs analyzed in the two systems (i.e. single plasmid-based M.HpaII DPCs in egg extracts vs. multiple DNMT1 DPCs distributed across human chromosomes), and the redundancy between different SUMO E3s in promoting DNMT1 DPC SUMOylation in cells seems well aligned with the overall strong magnitude of this response. We now mention and discuss this in the manuscript (pages 10 and 19).

As pointed out by the reviewer, one of the PIAS4 siRNAs used in the original manuscript gave rise to a modest decrease in 5-azadC-induced DNMT1 ubiquitylation (previous Fig. S2G). We observed such an effect for some PIAS4 siRNAs but not others (the underlying reason for this is not clear), arguing against this being a specific consequence of PIAS4 knockdown. In the revised manuscript, we included data showing the impact on DNMT1 DPC SUMOylation and ubiquitylation of PIAS4 knockdown by two independent siRNAs whose effects are overall representative of the range of PIAS4 siRNAs that we tested (new Fig. EV2K,L).

(3) The notion that unresolved DPCs impair mitosis is novel, but it was largely derived from experiments on DNMT1 DPCs. It is important to know whether the effect of DPCs on mitosis is common among various DPCs. The authors briefly addressed this by showing hypersensitivities of RNF4 knock down cells to formaldehyde, but such experiments might not necessarily reflect the role of RNF4 in DPC removal given that formaldehyde causes other types of DNA damage and RNF4 is widely involved in DNA damage response. The role of RNF4 in the clearance of formaldehyde-induced DPCs needs to be demonstrated, and then the impact of such DPCs on mitosis should be examined by treating RNF4-depleted G2 cells with physiologically relevant concentrations of formaldehyde.

We agree with the referee that extending studies of the adverse impact of DNMT-type DPCs on mitosis to other DPCs that are not accompanied by DNA breakage is of key importance for establishing whether or not this is a general effect of this class of lesions. Accordingly, as suggested by the reviewer, we performed a range of new experiments to carefully analyze whether the repair mechanisms for DNMT1 DPCs and the impact of blocking their resolution on mitotic progression and genome stability also apply to formaldehyde-induced DPCs. First, we found that exposing cells to a dose of formaldehyde that potently induces DPC formation and an accompanying dramatic chromatin SUMOylation response (Borgermann et al., EMBO J. 38:e101496 (2019)) has little impact on canonical DNA damage signaling (new Fig. EV4F), similar to the effect of 5-azadC-induced DPCs. Second, consistent with the hypersensitivity of RNF4-depleted cells to formaldehyde (Fig. EV5D), we show that RNF4 undergoes strong recruitment to chromatin upon DPC formation by formaldehyde treatment in a SUMOylation-dependent manner and is needed for efficient resolution of the resulting chromatin-associated SUMO foci (new Fig. EV1I,J). Third, using live-cell imaging of cells exposed to formaldehyde treatment in G2 phase to avoid adverse impacts on DNA replication, we show that suppressing the resolution of formaldehyde-induced DPCs via SUMOi treatment delays progression through mitosis and greatly enhances the incidence

of defective chromosome segregation (new Fig. EV5B,C), again paralleling the response to 5-azadC-induced DPCs. Collectively, these new data demonstrate that RNF4 also has a role in processing formaldehyde-induced DPCs and that these lesions do not trigger G2 cell cycle arrest but undermine mitotic fidelity if not properly resolved, strongly suggesting that this is a general effect of unresolved DPCs in uninterrupted duplex DNA.

Minor points:

Fig. 2I: The figure indicates U2OS was used but the legend describes HeLa/GFP-DNMT1. If U2OS was used, PIAS4 Western blotting should be shown. In addition, the authors need to indicate which siPIAS4 (#1 or #2) was used in Fig. 2I.

We apologize for the confusion. Careful analysis of the impact of a range of PIAS4 siRNAs on DPC repair showed that they do not consistently give rise to a pronounced delay in DNMT1 DPC resolution kinetics, consistent with PIAS4 being overall dispensable for 5-azadC-induced DNMT1 SUMOylation (new Fig. 2K; Fig. EV2K,L). We therefore removed the data originally shown in Fig. 2I from the manuscript. The specific PIAS4 siRNAs used in individual experiments are now indicated in the figures, and blots showing their knockdown efficiency in the relevant cell line have been added (new Fig. EV2L).

Thank you for submitting your revised manuscript to The EMBO Journal. We have now heard back from the two of the original referees, and I am pleased to say that they found the previously-raised points satisfactorily addressed. Following a final revision round to address some remaining minor issues noted by reviewer 3, as well as the below-listed editorial points, we shall therefore be happy to accept the study for publication in our journal.

REFEREE REPORTS

Referee #2:

The authors have performed many additional experiments and made a major effort to improve the manuscript. In addition, they also clarified the issues I suggested in the revised version of the manuscript (or in their answer to my questions). I have no further comments and gladly recommend publication of this work in EMBO journal.

Referee #3:

In the revised manuscript, the authors included new data demonstrating PIAS4 recruitment to DNMT1-DPCs and increased interaction of PIAS4 with DNMT1 after 5-azadC treatment, strengthening the notion that the SUMO pathway is activated locally at DNMT1-DPCs. A possible explanation for the weak effect of PIAS4 depletion is discussed properly.

The authors also provided new data demonstrating SUMO-dependent recruitment of RNF4 to the chromatin in response to formaldehyde treatment. In addition, they showed that SUMO inhibition caused mitotic defects after formaldehyde treatment. These data strengthened their notion that the SUMO-RNF4 pathway might be a general cellular response to a class of DPCs and that this mechanism is important for preventing mitotic defects due to post-replicative DPCs.

I believe that these revisions improved the manuscript significantly. If the authors address the concerns related to the newly added data (explained below), the manuscript would be suitable for publication in EMBO J.

Major point:

1. Fig. EV1J: It is not clear what the Y-axis shows. The axis title indicates "rel. to FA-treated" but does that mean the values are relative to the 0 hr time point? In that case, how exactly were the presented values calculated (the median of the 0 hr sample does not seem to be 1)?
2. Fig. EV1J: If the Y-axis is the relative values, why do they need to present data that way? The authors should simply plot the actual numbers of SUMO foci in each sample. The problem of

comparing relative values between siCTRL and siRNF4 is that it is not clear whether the same amount of SUMO foci was observed at the 0 hr time point. If the foci are lower in siCTRL cells for some reason, it is difficult to assess whether RNF4 plays an important role in dissolving formaldehyde-induced DPCs, because there might be simply fewer DPCs to repair in the siCTRL cells, which could provide an alternative explanation for the faster resolution of SUMO foci.

3. Fig. 2I: What is the molecular basis of the SUMO-dependence of PIAS4 recruitment? While the SUMO-dependent recruitment of RNF4 was explained in the manuscript, it is not immediately clear why PIAS4 recruitment should be SUMO-dependent as well. This might need more explanation to fully appreciate its significance.

Minor point:

1. Fig. 2J: Equal expression of Myc-PIAS4 is not demonstrated. A blot for Myc-PIAS4 should be included in the input.

Point-by-point reply to the referees' comments**Referee #3:**

In the revised manuscript, the authors included new data demonstrating PIAS4 recruitment to DNMT1-DPCs and increased interaction of PIAS4 with DNMT1 after 5-azadC treatment, strengthening the notion that the SUMO pathway is activated locally at DNMT1-DPCs. A possible explanation for the weak effect of PIAS4 depletion is discussed properly.

The authors also provided new data demonstrating SUMO-dependent recruitment of RNF4 to the chromatin in response to formaldehyde treatment. In addition, they showed that SUMO inhibition caused mitotic defects after formaldehyde treatment. These data strengthened their notion that the SUMO-RNF4 pathway might be a general cellular response to a class of DPCs and that this mechanism is important for preventing mitotic defects due to post-replicative DPCs.

I believe that these revisions improved the manuscript significantly. If the authors address the concerns related to the newly added data (explained below), the manuscript would be suitable for publication in EMBO J.

Major point:

1. Fig. EV1J: It is not clear what the Y-axis shows. The axis title indicates "rel. to FA-treated" but does that mean the values are relative to the 0 hr time point? In that case, how exactly were the presented values calculated (the median of the 0 hr sample does not seem to be 1)?

We apologize for the confusion. The data in this panel was represented as the number of SUMO foci/cell relative to cells exposed to a 1-hour formaldehyde pulse (recovery time point 0), but to avoid ambiguity we now display the data as SUMO foci/cell relative to untreated cells (Fig. EV1J). All data points were normalized to the mean of SUMO foci counts in the reference condition, which is why the median of this sample was not 1.

Please see also our response to the point below.

2. Fig. EV1J: If the Y-axis is the relative values, why do they need to present data that way? The authors should simply plot the actual numbers of SUMO foci in each sample. The problem of comparing relative values between siCTRL and siRNF4 is that it is not clear whether the same amount of SUMO foci was observed at the 0 hr time point. If the foci are lower in siCTRL cells for some reason, it is difficult to assess whether RNF4 plays an important role in dissolving formaldehyde-induced DPCs, because there might be simply fewer DPCs to repair in the siCTRL cells, which could provide an alternative explanation for the faster resolution of SUMO foci.

The reason we chose to present the data in Fig. EV1J as relative values is that we consistently observed across multiple experiments that RNF4-depleted cells accrue a higher number of SUMO foci following formaldehyde treatment relative to control cells (see new Fig. EV1K), in full agreement with an important role of RNF4 in counteracting formaldehyde-induced DPCs. Displaying the data as relative values in our opinion facilitates direct comparison of the rate of SUMO foci resolution in control and RNF4-depleted cells. However, to also convey the important point that formaldehyde-induced SUMO foci accumulate to a higher level in RNF4-depleted cells, we now additionally include a plot of the data in Fig. EV1J that depicts the total number of SUMO foci (new Fig. EV1K). Reflecting the reviewer's concern that the lower number of SUMO foci in

control cells could be a contributing factor to their faster resolution relative to cells lacking RNF4, which we cannot formally exclude, we rephrased the sentence describing this data so that it now reads: “Moreover, RNF4 was required for counteracting chromatin-associated SUMO2/3 foci resulting from formaldehyde-induced DPC formation (Fig. EV1J,K)” (page 9).

3. Fig. 2I: What is the molecular basis of the SUMO-dependence of PIAS4 recruitment? While the SUMO-dependent recruitment of RNF4 was explained in the manuscript, it is not immediately clear why PIAS4 recruitment should be SUMO-dependent as well. This might need more explanation to fully appreciate its significance.

We agree with the reviewer that this point was not explained well in the revised manuscript. The SUMO-dependent recruitment of PIAS4 to 5-azadC-induced DPCs in human cells (Fig. 2I) is consistent with our findings in *Xenopus* egg extracts showing SUMO-dependent chromatin accumulation of PIAS4 upon PARP1 trapping (Fig. 2E; new Fig. EV2B) in a manner that depends on its dual SUMO-interacting motifs (SIMs) (Fig. EV2D,E; new Fig. EV2G); we now clarify this as follows in the text: “In cells, consistent with our observations in *Xenopus* egg extracts, PIAS4 was recruited to DNMT1 DPC sites in a SUMOylation-dependent manner and displayed increased binding to DNMT1 upon DPC formation (Fig. 2I,J; Fig. EV2K)...” (page 11). Interestingly, further analysis of the SUMOylation-dependent accumulation of PIAS4 at DNMT1 DPC sites revealed that this does not require the catalytic activity of PIAS4 itself, as an ectopically expressed inactive PIAS4 mutant displayed intact recruitment to DPCs even when endogenous PIAS4 was simultaneously depleted (new Fig. EV2M,N). This suggests a role for one or more other SUMO E3 ligases in driving PIAS4 accumulation at DPC sites, in good agreement with our data indicating redundancy between PIAS4 and other SUMO E3s in promoting DNMT1 DPC SUMOylation in human cells (Fig. 2K; Fig. EV2K,L).

Minor point:

1. Fig. 2J: Equal expression of Myc-PIAS4 is not demonstrated. A blot for Myc-PIAS4 should be included in the input.

In this experiment, individual GFP immunoprecipitates were incubated with an equal amount of whole cell lysate prepared from a single plate of HeLa cells transfected with Myc-PIAS4 expression construct. We clarified this in the legend and added an immunoblot showing Myc-PIAS4 expression in this lysate (new Fig. EV2J).

Thank you for submitting your final revised manuscript for our consideration. I am pleased to inform you that we have now accepted it for publication in The EMBO Journal.

Corresponding Author Name: Niels Mailand

Journal Submitted to: The EMBO Journal

Manuscript Number: EMBOJ-2020-107413